# Effect of the Inlet Boundary Conditions on the Flow over Complex Terrain Using Large Eddy Simulation

**Yi Wang** [1,2] , **Giulio Vita** [1] , **Bruño Fraga** [1] , **Jianchun Wang** [2] **and Hassan Hemida** [1,*]

1 School of Civil Engineering, University of Birmingham, Birmingham B15 2TT, UK;
11756010@mail.sustech.edu.cn (Y.W.); G.Vita@bham.ac.uk (G.V.); B.Fraga@bham.ac.uk (B.F.)
2 Department of Mechanics and Aerospace Engineering, Southern University of Science and Technology,
Shenzhen 518055, China; wangjc@sustech.edu.cn
* Correspondence: H.Hemida@bham.ac.uk

**Abstract:** For large eddy simulation, it is critical to choose the suitable turbulent inlet boundary condition as it significantly affects the calculated flow field. In this paper, the effect of different inlet boundary conditions, including random method (RAND), Lund method, and divergence-free synthetic eddies method (DFSEM), on the flow in a channel with a hump are investigated through large-eddy simulation. The simulation results are further compared with experimental data. It has been found that turbulence is nearly fully developed in the case based on the Lund method, not fully developed in the case based on DFSEM, and not developed in the case based on the RAND method. In the flow region before the hump, mean velocity profiles in the case applying the Lund method gradually fit the law of the wall as the main flow moves towards the hump, but the simulation results based on the RAND and DFSEM methods cannot fit the wall function. In the flow region after the hump, cases applying Lund and DFSEM methods could relative precisely predict the size of turbulent bubble and turbulent statistics profiles. Meanwhile, the case based on the RAND method cannot capture the positions of flow separation and re-attachment point and overestimates the turbulent bubble size. From this research, it could be found that different turbulent inflow generation methods have a manifested impact on the flow separation and re-attachment after the hump. If the coherent turbulence is maintained in the approach flow, even though turbulent intensity is not large enough, the simulation can still predict the flow separation and turbulent bubble size relative precisely.

**Keywords:** computational fluid dynamics; large eddy simulation; Inlet boundary condition

## 1. Introduction

Atmospheric wind is almost always turbulent and has a strong impact on the aerodynamic loads for objects in the atmospheric boundary layer (ABL), such as wind turbines, ground transportation, high-rise buildings, and many other engineering applications. Especially in complex terrain, local wind phenomena can be expected to greatly affect the aerodynamic loads on these applications [1]. The inherent unsteady features of the flow must be simulated to provide reliable predictions. To accurately predict the effects of turbulent flow on the performance, loads and reliability of such applications, it is significant to calculate the properties of atmospheric flow close to the ground [2]. The atmospheric boundary layer has been investigated through experiments [3] and computational fluid dynamics (CFD) simulations.

CFD can be conducted through different simulation methods, including direct numerical simulation (DNS), large-eddy simulation (LES) and Reynolds Averaged Navier-Stokes (RANS). DNS directly resolve all the non-linear mechanisms of turbulence production and the viscous dissipation. However, DNS cannot be applied to high Reynolds number atmospheric flows since the associated computational cost increases with the Reynolds number as $Re^3$ [4]. In contrast to DNS, a RANS simulation extracts the steady solution for the mean flow field. Most predictions for engineering applications are obtained from solution of the

RANS equations [4]. The RANS model can predict the attached flows and some shallow separation flows. For instance, RANS can be used to investigate the average quantities of ABL. Despite this usefulness, RANS methods technically fail to predict complex flow structures and it cannot capture the instantaneous properties close to the ground [5]. More recently, large eddy simulation (LES) of the simulation method has become an alternative to predict the flow field [6].

In LES, the larger three-dimensional unsteady turbulent motions are directly resolved, whereas the effects of the smaller-scale motions are modelled. In terms of computational expense, LES lies between RANS and DNS and it is motivated by the limitations of each of these approaches [7]. In LES, the grid scale variables always contain some temporary components, stochastically varying on all scales down to the smallest spatial and temporary scales of the simulation. These properties make the results strongly influenced by inlet boundary conditions (IBC) including the mean velocity profiles and turbulent intensity [8]. In order to obtain fully developed turbulent flow without significantly increasing computation, it is necessary to implement reliable turbulence IBC [9]. Currently, there are several methods capable of generating IBCs and these methods can be classified into two classes: precursor simulation methods and synthetic methods [8].

Precursor simulation methods produce inflow turbulence by using CFD without reliance on artificial turbulent signals. The most straightforward approach consists of two steps. Firstly, it starts from the easy-to-specify laminar regime including disturbances and simulating transition to turbulence. Then, it uses this information as input to the main domain of interest [10]. However, the main drawback of this method is the high computational cost due to the need for a very long computational domain and the implementation limitations regarding the highest achievable Reynolds number [11]. Besides, the precursor simulation with recycling and rescaling process called Lund method has been put in practice for modelling turbulent inflow conditions. A comprehensive review can be found in the work of Lund et al. [12], Moin and Mahesh [13], and Keating et al. [14]. This method applied in the zero-pressure gradient (ZPG) flat plate boundary layer with $Re_\theta = 1410$ performs well in the Spalart case [12]. Nevertheless, the technique is limited in the principle to ZPG equilibrium boundary layers. The limitation arises because it is assumed that the inner and outer regions have a single velocity scale, and an empirical correlation needs to be specified to connect the friction velocities $u_\tau$ between the inlet and recycle planes. Araya et al. prescribe inlet turbulent conditions, which addresses the limitations of the Lund method, and applied their method in the flat plate boundary layer with favourable and adverse pressure gradients [15].

For the synthetized turbulence methods, turbulence is generated by superimposing artificially generated fluctuations on the statistically averaged properties [16]. The simplest method is the random method (RAND), which generates turbulence at the inlet by adding random noise based on the value of the average velocity [17]. The RAND method results in a uniform energy distribution of high-frequency and low-frequency turbulence. Since high-frequency turbulence disappears in a short term, this will produce unstable turbulence. To improve the RAND method, the langevin-type equations were used which can provide coherent fluctuations [18]. In contrast to the mere Gaussian fluctuations, the Langevin-type method can provide enough coherence for the turbulence not to be damped. However, the method cannot provide the spatial correlation of fluctuating velocity among the neighbour cells. Fourier approach is another branch of synthetic turbulence method. Smirnov [19] proposed this approach and developed it based on the work of Kraichnan [20]. If the anisotropic velocity correlation tensor is given, the method can generate isotropic, divergence-free fluctuating velocity fields that satisfy the Gaussian spectrum model, as well as non-uniform anisotropic turbulence. Huang et al. [21] proposed the random flow generation (DSRFG) method that can make the spatially correlated turbulent flow satisfy any arbitrary model spectrum. This property is useful in computational wind engineering applications where the von Karman model is widely adopted as a target spectrum and the energy content of the inertial subrange cannot be discarded. It has been proved that the

DSRFG method can improve the accuracy of turbulence simulation and wind direction on buildings, but there are few discussions about the time correlation of the synthetic turbulence generated by this method. Castro et al. [9] proposed a modified DSRFG method, it preserves the statistical quantities that would be prescribed at the inlet of the domain independently of the number of points in the spectrum. Furthermore, the generation of each nodal fluctuating velocity component can be done prior to computation by LES using DSRFG, calling in each time step the corresponding nodal value.

Besides the Fourier approach, the synthetic eddies method (SEM) [22] is used to generate inflow turbulence as well. This method was based on a three-dimensional correlation of fluctuations with a predefined shape function and demonstrated an improved downstream development compared to other formulations. However, the turbulence velocity field simulated with the SEM with single length scale eddies can hardly meet the spectrum of velocity fluctuation. Luo et al. [23] developed this method consisting of multi-scale eddies with different length scales as a substitute of the single-scale eddies and named it as multi-scale synthetic eddies method (MSSEM). This method not only inherits the advantages of SEM but also prescribes both coherent structures and turbulence spectra. However, the above schemes are that the fluctuating velocity fields they produce are not generally divergence-free. If the inlet velocity is not from a non-divergence field, it may cause large pressure fluctuations near the inlet, resulting in the required rapid velocity change. Poletto et al. [24] proposed a divergence free synthetic eddy method (DFSEM) to overcome this problem. The digital filter developed by Klein et al. is another general approach in the branch of synthetized turbulence methods. The technique was based on the knowledge that for late-stage homogeneous turbulence, the correlation function takes a Gaussian form and a three-dimensional digital filter is used [17]. Xie et al. [25] developed the digital method on the basis of the Klein's work, and this method allows spatially varying turbulence scales on non-uniform grids to be imposed at the inlet based on exponential (rather than Gaussian) velocity correlation functions. Moreover, this method is used only for the generation of spatially correlated two-dimensional slices of data with a two-dimensional filter [26].

The channel flow with a hump is seen as a flow on an isolated hill, which is a simplified model for simulating ABL in complex terrain [27]. Even though other work has investigated the effect of all IBCs on LES results for the flat plate boundary layer [24–29], all IBCs that were not applied to the channel flow over a hump discussed here [29]. Moreover, previous simulations consider more about the influence of IBCs on LES for the flow upstream the hump, there is little research focusing on the effects of IBCs on the flow separation after the hump [30]. Flow separation and reattachment phenomena are accompanied by a substantial loss of energy, affecting the performance of the fluid machine, and severely limiting the design and operation of many of the fluid flow devices [31].

In this paper, the effects of using different IBCs in LES on the flow over a hump are investigated. The flow properties, such as mean velocity profiles, Reynolds stresses profiles, and pressure coefficients distribution in the approach flow as well as the flow downstream were calculated through applying different IBCs. The spanwise domain size sensitivity analysis is not carried out in this work that should be done in the future. Applicability of different methods in LES for various parameters is discussed and the results of these methods are compared with the data obtained experimentally. The work provides potential strategies for implementation of IBC methods on the flat plate boundary layer, complex terrain problems and then the ABL when using LES as the simulation method [32].

## 2. Numerical Method

In this section, the specific conditions of LES applied are introduced. The filtered incompressible governing equations are:

$$\frac{\partial \overline{u}_i}{\partial t} + \frac{\partial}{\partial x_j}\left(\overline{u}_i \overline{u}_j\right) = -\frac{\partial \overline{p}}{\partial x_j} + \frac{\partial}{\partial x_j}\left(\frac{\partial \overline{u}_i}{\partial x_j} + \frac{\partial \overline{u}_j}{\partial x_i}\right) - \frac{\partial \tau_{ij}}{\partial x_j} \tag{1}$$

$$\frac{\partial \overline{u}_i}{\partial x_i} = 0 \tag{2}$$

The overbar represents filtered quantities. The fluctuating component, $u'$, omitted due to the filtering process, can be calculated by:

$$u' = u - \overline{u} \tag{3}$$

The unresolved sub-grid stresses (SGS), $\tau_{ij}$ in Equation (1) can be calculated by different SGS models. The turbulence unresolved in the Kolmogorov length scale can be regarded as fully developed isotropic homogeneous turbulence. SGS stresses are given by:

$$\tau_{ij} - \frac{1}{3}\delta_{ij}\tau_{kk} = -2v_T\overline{S}_{ij} = -2C\Delta^2|\overline{S}|\overline{S}_{ij} \tag{4}$$

where $\overline{S}_{ij}$ is called the "resolved strain rate", $\overline{S}_{ij} = 1/2(\partial \overline{u}_i/\partial x_j + \partial \overline{u}_j/\partial x_i)$, and $v_T$ is the Smagorinsky eddy viscosity, which can be represented as:

$$v_T = (C_s\Delta)^2\sqrt{\overline{S}_{ij}\,\overline{S}_{ij}} \tag{5}$$

where $C_s$ is the Smagorinsky coefficient. The values of $C_s$ should be adjusted to get the best results on different flow conditions. Equation (5) is akin to a mixing-length formula with mixing length of $C_s\Delta$. Here, $\Delta$ is the grid scale, which equals $(\Delta_1\Delta_2\Delta_3)^{1/3}$. This model is stable and robust because it yields enough diffusion and dissipation to the numerical computations. The Smagorinsky model does not perform well if the mesh close to the boundary is too coarse. Therefore, a damping function $f_\mu$ is induced in this case, [33]

$$f_\mu = 1 - exp(-y^+/26), \; v_T = (C_s f_\mu \Delta)^2\sqrt{2\overline{S}_{ij}\,\overline{S}_{ij}} \tag{6}$$

## 3. Overview of Inflow Boundary Condition Methods

### 3.1. Recycling and Rescaling Method (Lund)

The Lund method is an approach to extract instantaneous velocity data at a fixed plane in an auxiliary simulation. The auxiliary simulation is spatially developing but generates its own inflow conditions through a sequence of operations where the velocity field at a downstream station is rescaled and re-introduced at the inlet, this process is illustrated to Figure 1. The rescaling factors are determined by the parameters of turbulent boundary layer thickness, displacement thickness and the momentum thickness at the beginning. The velocity field with random noise is generated first, and the mean velocity profile is derived from the value of the displacement thickness [15].

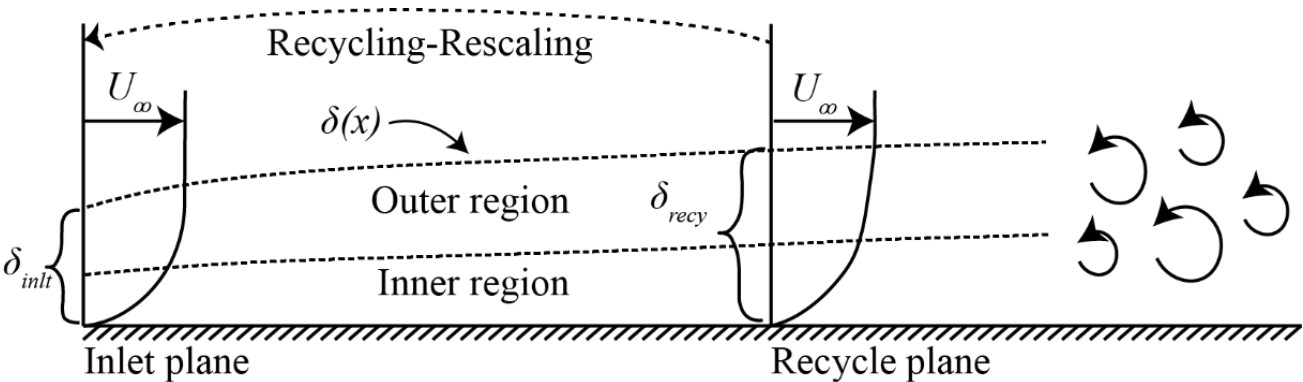

**Figure 1.** Simulation of the recycling and rescaling method.

For the inlet plane and the recycled plane, the mean velocity meets that

$$U_{inlt}^{inner} \big/ U_{recy}^{inner} \left(y_{inlt}^{+}\right) = \left(U^{\infty} - U_{inlt}^{outer}\right) \big/ \left(U^{\infty} - U_{recy}^{outer}\left(\eta_{inlt}\right)\right) = u_{\tau}^{inlt} \big/ u_{\tau}^{recy} = \gamma \quad (7)$$

and the fluctuating velocity is

$$\left(u_i'\right)_{inlt}^{inner} \big/ \left(u_i'\right)_{recy}\left(y_{inlt}^{+}\right) = \left(u_i'\right)_{inlt}^{outer} \big/ \left(u_i'\right)_{recy}\left(\eta_{inlt}\right) = u_{\tau}^{inlt} \big/ u_{\tau}^{recy} = \gamma \quad (8)$$

where $y^+$ and $\eta$ are defined at the inlet, and $y^+ = \frac{u_\tau y}{v}, \eta = \frac{y}{\delta}$, where $\delta$ is the boundary layer thickness determined at the inlet plane. The velocity profiles in the inner and the outer region can be combined with a weight function,

$$\left(u_i\right)_{inlt} = \left[\left(U_i\right)_{inlt}^{inner} + \left(u_i'\right)_{inlt}^{inner}\right]\left[1 - W(\eta_{inlt})\right] + \left[\left(U_i\right)_{inlt}^{outer} + \left(u_i'\right)_{inlt}^{outer}\right]W(\eta_{inlt}) \quad (9)$$

the weighting function $W(\eta)$ is defined as:

$$W(\eta) = \frac{1}{2}\left\{1 + tanh\left[\frac{\alpha(\eta - b)}{(1 - 2b)\eta + b}\right] \big/ tanh(\alpha)\right\} \quad (10)$$

where $\alpha = 0.4$ and $b = 0.2$. For $\alpha \longrightarrow \infty$, $W(\eta)$ becomes a step function centred at $\eta = b$. As $\alpha \longrightarrow 0$, the transition is spread across the entire boundary layer. The values of $\alpha$ and $b$ are determined from the analysis of the independent spatially evolving boundary layer. The rescaling operation requires the scaling parameters $u_\tau$ and $\delta$ at the recycle station at the inlet. There is a suitable relation where:

$$\frac{u_{\tau, \, inlt}}{u_{\tau, \, resc}} = \left(\frac{\theta_{recy}}{\theta_{inlt}}\right)^{1/[2(n-1)]} = \gamma \quad (11)$$

where $\theta_{inlt}$ is the momentum thickness at the inlet and $\theta_{recy}$ is the momentum thickness at the recycling plane. In many cases, it is more advantageous to control the inlet momentum thickness than the inlet boundary layer thickness. This can be done with a little extra effort by iteratively adjusting the inlet boundary layer thickness until the target inlet momentum thickness is achieved.

The time average used to compute the mean velocity field is a simple running average when the flow is fully developed, but it should be modified in order to eliminate the starting transients if the solution is initialized with a crude guess. A convenient way to realize this process by using the formula,

$$U^{n+1} = \frac{\Delta t}{T}\left\langle u^{n+1}\right\rangle_z + \left(1 - \frac{\Delta t}{T}\right)U^n \quad (12)$$

where $\Delta t$ is the computational time step, $T$ is the characteristic time scale of the averaging interval, and $\langle\rangle_z$ denotes an average in the spanwise direction.

### 3.2. Random Method (RAND)

The most straightforward method to generate the turbulence at the inlet is to randomize noise on the basis of the mean velocity,

$$u_i^n = (1 - \alpha)u_i^{n-1} + \alpha(U + sRU) \quad (13)$$

In Equation (13) $u_i^n$ is the current instantaneous velocity, and $u_i^{n-1}$ is the previous instantaneous velocity. $U$ is spatial and temporal average velocity. $R$ is the random value subject to standard normal distribution. $\alpha$ is the weighting average factor and s is the fluctuation scale [34].

### 3.3. Divergence Free Synthetic Eddies Method (DFSEM)

The DFSEM is based on the methodology described in [22,24]. In this method, the synthetic eddies represent a set of velocity fluctuations in the vicinity of the inlet plane where a turbulent field is required. These eddies are defined by their centres and the formula of the velocity distribution around these centres. All these synthetic eddies and the inlet plane are encompassed by a virtual box. Eddies generated at the head of the virtual box are convected by mean flow at each time step. Then, they transverse the inlet plane encircled by the virtual box and finally leave the box. When they leave the box, they will regenerate anywhere on the head of the box and repeat the process described above. Instantaneous velocity information is recorded at the inlet plane. In general, the steps of the DFSEM algorithm are summarized by a flow chart shown in Figure 2.

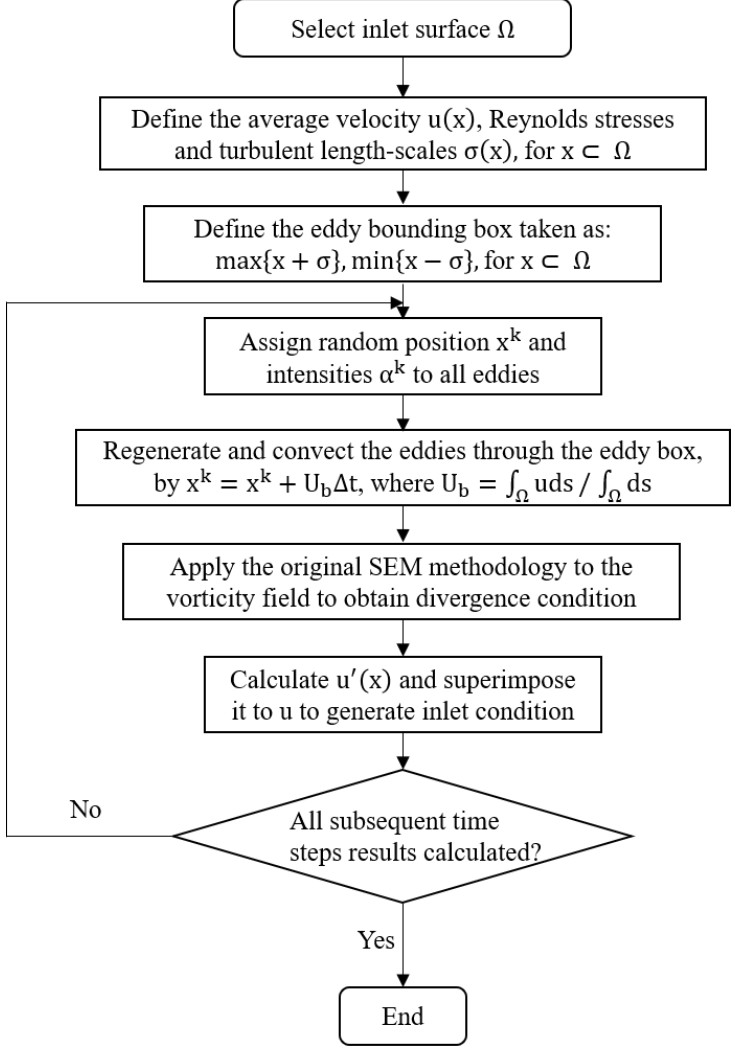

**Figure 2.** Flow chart of the application of Divergence Free Synthetic Eddies Method (DFSEM).

The DFSEM defines velocity fluctuations, $u_i'(x)$, as:

$$u_i'(x) = \frac{1}{\sqrt{N}} \sum_{K=1}^{N} a_{ij}\varepsilon_j^k f_\sigma^k \left( \frac{x - x^k}{\sigma^k} \right) \tag{14}$$

where $N$ is the number of eddies introduced into the SEM domain; $x^k$ is the location of the centre of the kth eddy; $\sigma^k$ is the turbulence length-scale of eddies; $f_\sigma(x)$ is a suitable

shape function; $\varepsilon_j^k$ are random numbers with zero average and $\left\langle \varepsilon_j^k \varepsilon_j^k \right\rangle = 1$. $a_{ij}$ represents the eddy intensities, and written as:

$$a_{ij} = \begin{bmatrix} \sqrt{R_{11}} & 0 & 0 \\ \frac{R_{21}}{a_{11}} & \sqrt{R_{22} - a_{21}^2} & 0 \\ \frac{R_{31}}{a_{11}} & \frac{R_{32} - a_{22}a_{31}}{a_{22}} & \sqrt{R_{33} - a_{31}^2 - a_{32}^2} \end{bmatrix} \tag{15}$$

where $R_{ij}$ are the elements of the Reynolds stresses tensor. Although this formulation allows any desired Reynolds stress field to be prescribed (via the $a_{ij}$ coefficients), the velocity field will not be divergence-free. One route to obtain a divergence-free method is to apply the original SEM methodology to the vorticity field, which is then transformed back to the velocity field by taking the curl of it.

$$\nabla \times \omega' = \nabla \left( \nabla \cdot u' \right) - \nabla^2 u' \tag{16}$$

where, because of the hypothesis of incompressible flow, the first term on the right-hand side vanishes, leading to a Poisson equation, achieved by

$$u'(x) = \frac{1}{\sqrt{N}} \sum_{K=1}^{N} \frac{q_\sigma \left( \left| x^k \right| \right)}{\left| r^k \right|^3} r^k \times \alpha^k \tag{17}$$

where $r^k = \frac{x - x^k}{\sigma^k}$, $q_\sigma \left( \left| r^k \right| \right)$ is a suitable shape function and $\alpha_i^k$ are random numbers with zero average which represent the eddy intensities.

## 4. Numerical Setup

### 4.1. Hump Model

In this section, different IBCs implemented in a channel with a hump are examined and discussed. The chord length (c) of the hump is 0.42 m. The flow scenarios are considered under the condition of Mach number equals 0.1. This flow configuration illustrated in Figure 3a,b is one of the test cases considered in NASA Workshop [35], and experimental data can be accessed from the website of ERCOFTAC [36]. To generate the target characteristics of inflow turbulence, some essential properties of inflow turbulence are required. Boundary layer thickness, displacement thickness, and momentum thickness are pre-conditions of the Lund method. From the experimental data of ERCOFTAC, free-stream velocity $U_{inf}$ is 34.6 m·s$^{-1}$. The Reynolds number based on boundary layer thickness is $6.295 \times 10^4$. Figure 4 describes the hump model used in the simulation, the origin $x = 0$ is located on the beginning of the hump. The length of the hump chord $c = 0.42$ m, the channel height is $0.91c$ with a small necking over the hump to account for blockage effects, and the channel width is $0.2c$. Periodic boundary conditions have been set on the lateral faces due to the periodicity of the geometry in the spanwise direction. At the top surface, the values of velocity gradients and pressure are set to be zero. At the bottom wall, the wall function is used. In this case, we use the Spalding law wall function since it matches the asymptotic solutions in the inertial sublayer and is not highly affected by the values of $y+$ near the wall [37].

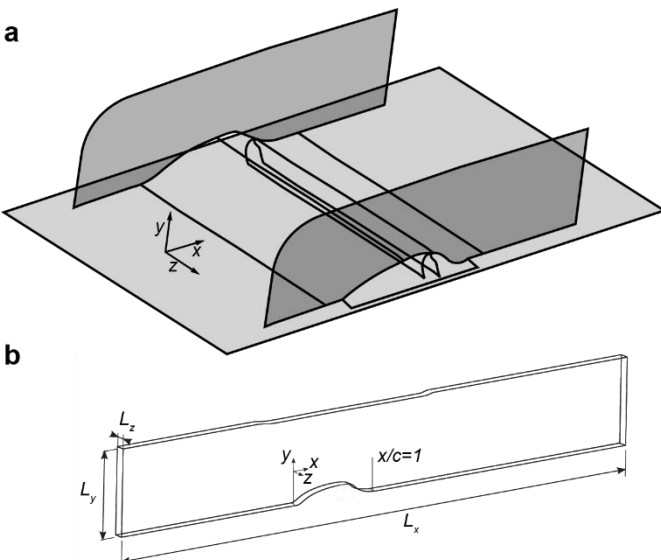

**Figure 3.** (**a**) Experimental model of the wall-mounted hump given by ERCOFTAC. (**b**) the simulation domain for hump case: $L_x = 6.14c$; $L_x = 6.14c$ ; $L_z = 0.2c$.

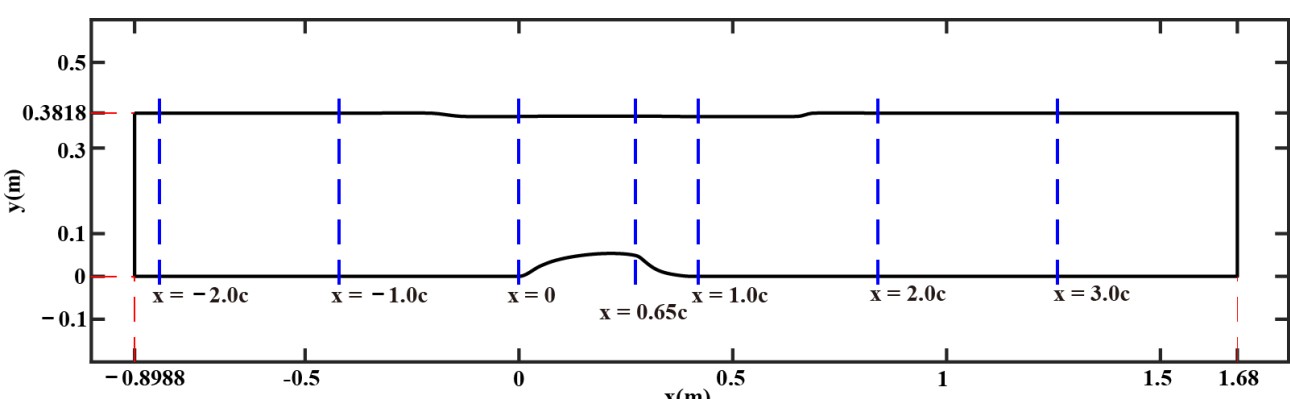

**Figure 4.** Side view of the computational domain with main dimensions expressed in chord unit length. Inlet is located at $x/c = -2.14$, while hump profile extends from $x/c = 0$ and $x/c = 1$. Outflow is at $x/c = 4$. Channel height is 0.909c. Chanel extends for 0.2c in $z$ direction.

$$Y_\tau^+ = U_1^+ + exp(-\kappa C)\left[exp\left(\kappa U_1^+\right) - 1 - \kappa U_1^+ - \frac{1}{2}\left(\kappa U_1^+\right)^2 - \frac{1}{6}\left(\kappa U_1^+\right)^3\right] \tag{18}$$

A backward differencing scheme has been used for the discretization of time derivatives. The convection term in momentum equation has been calculated using the Gauss theorem and a pure second order linear interpolation with non-orthogonal correction [30]. The divergence term in momentum has been calculated using the Gauss Linear-Upwind Stabilized Transport scheme. Other divergence terms have been discretized using Gauss linear schemes. The Courant–Friedrichs–Lewy (CFL) number in the simulation is always smaller than 1.

### 4.2. Different IBCs Used in the Simulation

This section elucidates detailed settings of simulations with different IBCs, these details are summarized in Tables 1 and 2, Figure 5. The specified thickness in Lund method including boundary layer thickness, momentum thickness and displacement thickness are required at the inlet, these properties can be calculated from the experimental data according to the Equations (7)–(11), where the boundary layer thickness $\delta$ is 0.0671c, displacement thickness $\delta^*$ is 0.009c, and momentum thickness $\delta_\theta$ is 0.0069c. With the

application of DFSEM, three mean velocity components, turbulent Reynolds stresses, and turbulent length scale at the inlet are needed. The initial setting details of the simulation based on DFSEM are illustrated in Figure 5.

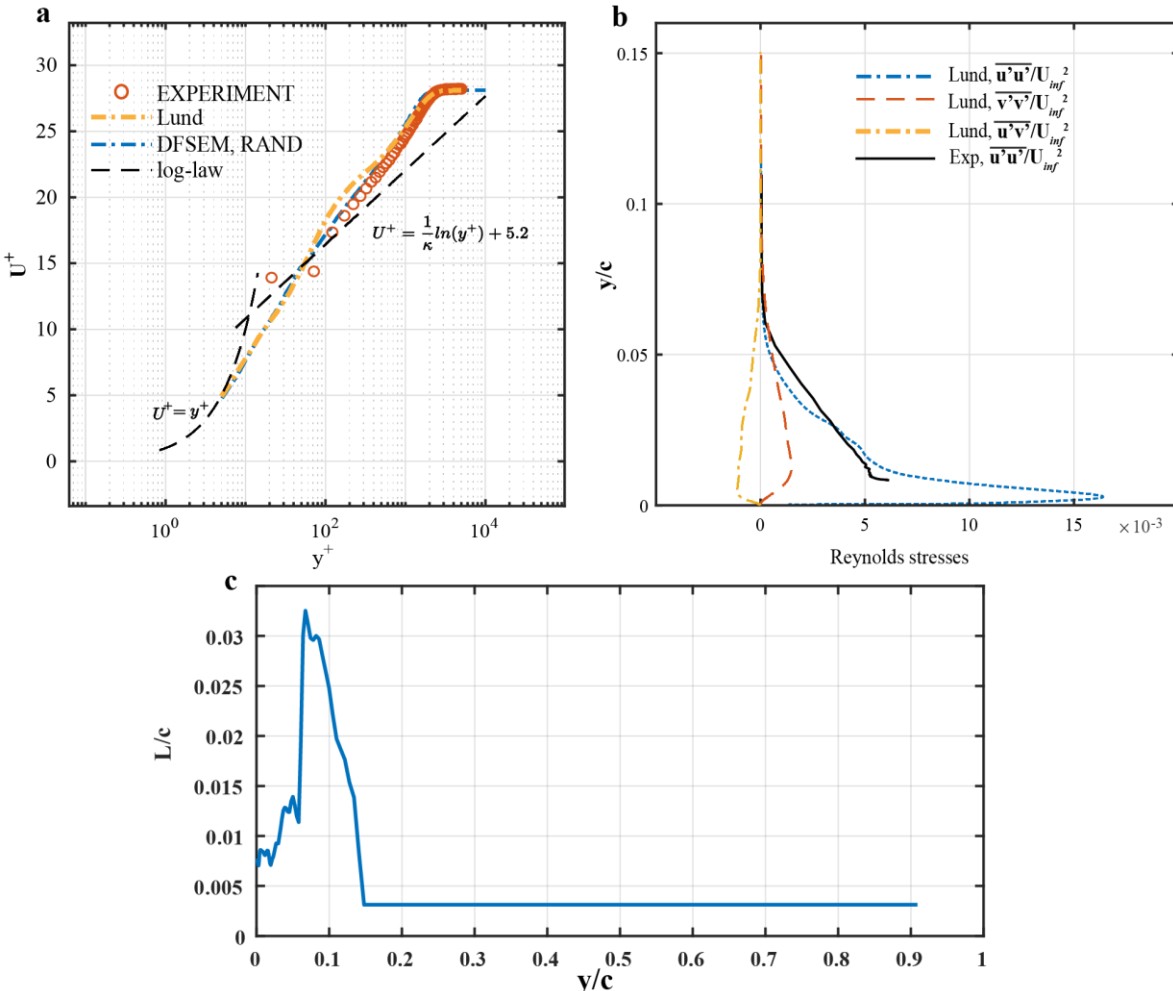

**Figure 5.** Input for the simulation based on DFSEM at the inlet (**a**) Nondimensional time-averaged velocity profiles at the inlet (**b**) Distribution of Reynolds stresses components $\overline{u'u'}, \overline{v'v'}$ and $\overline{u'v'}$ at the inlet of simulation based on Lund method and experimental data $(\overline{u'u'})$. (**c**) Profile of the turbulent length scale calculated from the simulation based on Lund method.

**Table 1.** Summary of the inflow boundary conditions.

|  | Mean Velocity | Specified Thicknesses | Reynold Stresses | Turbulent Length Scales |
|---|---|---|---|---|
| Random | X | — | — | — |
| Lund | X | X | — | — |
| DFSEM | X | — | X | X |

In Table 2, distribution of mesh cells along streamwise, vertical and spanwise directions is displayed. Three sets of mesh configuration are used to eliminate the influence of the mesh resolution on the simulation results. $\Delta x^+$, $\Delta y^+$, and $\Delta z^+$ are dimensionless mesh scale at the inlet plane. The mesh in the streamwise direction and vertical direction is not uniform, the mesh in the spanwise direction is uniform. The Smagorinsky model is used as the SGS model in each case and the value of $Cs$ is 0.1678.

**Table 2.** Summary of the mesh configurations.

| Density | Mesh Size (x, y, z) | $\Delta x^+$ | $\Delta y^+$ | $\Delta z^+$ |
|---------|---------------------|--------------|--------------|--------------|
| Coarse | $420 \times 88 \times 52$ | 929 | 6.35 | 125.06 |
| Middle | $540 \times 120 \times 64$ | 658 | 4.76 | 101.61 |
| Fine | $630 \times 140 \times 84$ | 619 | 3.18 | 77.42 |

*4.3. Inlet Profiles Validation of Different Simulation Methods*

Figure 5a shows the nondimensional time-averaged velocity profile for the experimental data and simulation results obtained by the implementation of three IBCs on the inlet boundary. The nondimensional time-averaged velocity profiles in the DFSEM and RAND methods are directly set by using the experimental data. In the case based on RAND method. Mean velocity $U$, weighting factor $\alpha$, and fluctuation scale s for each velocity component should be determined. In this simulation, the fluctuation scale $s$ for each velocity components in $x$, $y$, and $z$ direction are 0.02, 0.01 and 0.01 respectively. The weighting factor $\alpha$ is set to be 0.1. Mean velocity is determined from the interpolation of the experimental data as shown in Figure 5a.

The velocity profile generated by Lund method is derived from the recycling and rescaling process, the results of which are slightly different from the experimental data. It is barely to see large difference of the mean velocity profiles outside the turbulent boundary layer ($y^+ > 2300$) at the inlet among all simulation cases. Volume flow rates for all cases are consistent and equal to 1.1 m$^3$/s though there is small difference between the velocity profiles of different inlet boundary condition methods and experimental data in the near-wall region. However, the nondimensional time-averaged velocity profiles for all cases and the experiment data deviate from the theoretical velocity profile (log-law).

Figure 5b reflects the distribution of the Reynolds stresses components, $\overline{u'u'}$,$\overline{v'v'}$ and $\overline{u'v'}$ of the simulation based on the Lund method and $\overline{u'u'}$ distribution of experiment. From the graph, there is a small difference between the simulation results of Lund method and the experimental data close to the wall. The profiles of the Reynolds stresses generated by the Lund method at the inlet are also used as the initial setting for DFSEM to calculate the turbulent intensity at the inlet as explained in Equation (15). Figure 5c shows the profile of the turbulent length scale setting for DFSEM. The streamwise integral length scale $L_x$ is characteristic of the larger eddies, which is obtained through:

$$L_x \equiv \int_0^\infty \frac{\overline{u'(x)u'(x+r)}}{\overline{u'^2}} dr \tag{19}$$

where $r$ is the space lag in the spanwise direction. The turbulent length scale defined by the integral of autocorrelation function when it reaches *1/e* [38]. $u'(x)$ is fluctuating velocity components along the streamwise direction that is extracted from the prescribed simulation results based on the Lund method. Figure 5 illustrates the distribution of the streamwise integral length scale along the vertical direction. The values are used as the initial setting of DFSEM.

## 5. Results

*5.1. Mesh Sensitivity*

Figures 6–8 reflect the mesh sensitivity of the simulations based on Lund method, DFSEM and RAND method with different mesh resolution. The dimensionless friction velocity $u_\tau/U$ is calculated by the square root of wall shear stress at the inlet, which equal 0.035 and the value is fixed for each case since it can ensure that the magnitude of free stream velocity ($y^+ > 3000$) in the log-law plot is the same for each case. Differences of velocity profiles applying different mesh configurations are reflected by the velocity profiles in the near wall region for cases with different IBC methods.

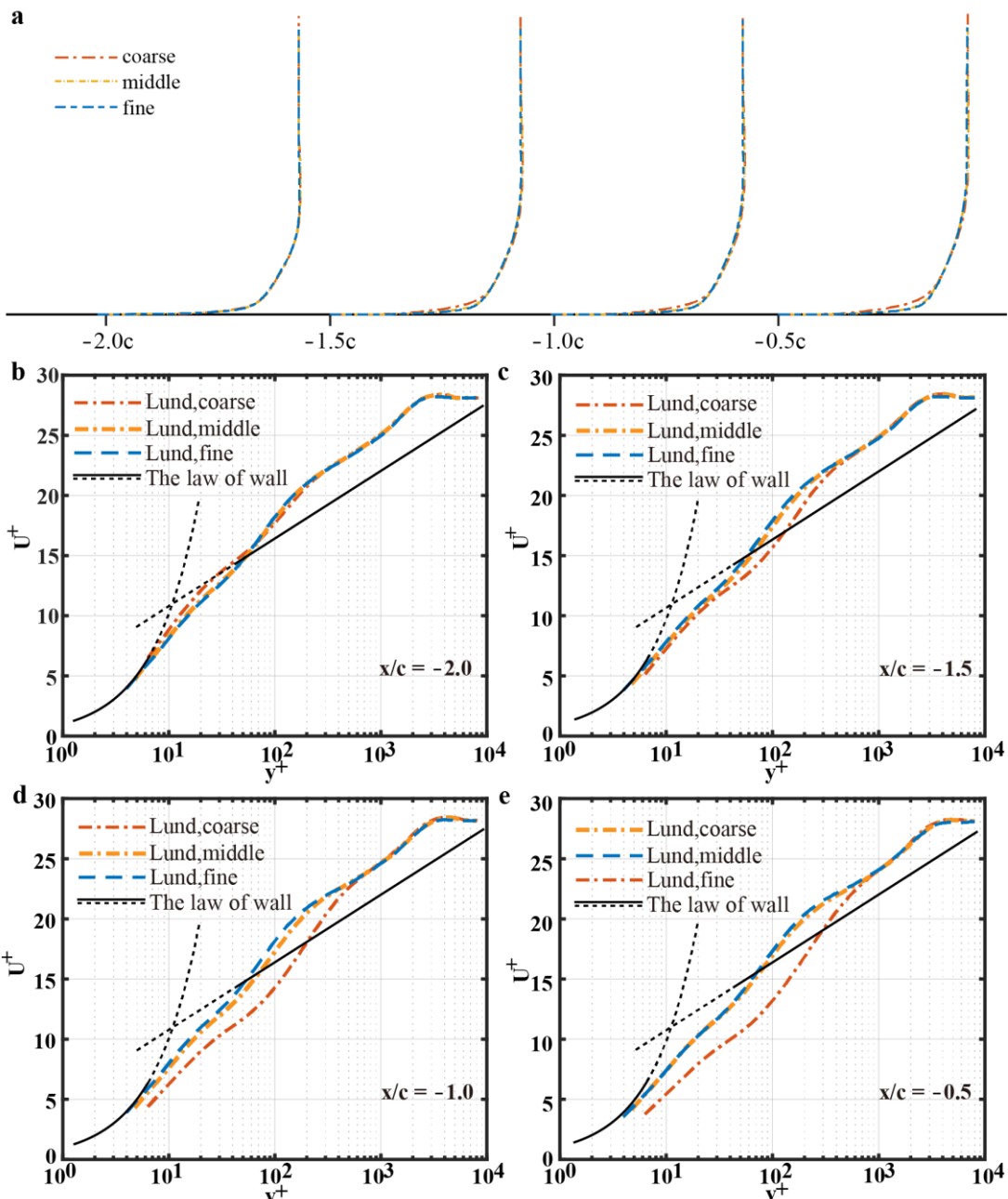

**Figure 6.** Effect of mesh resolution on streamwise mean velocity based on Lund method at (**a**) 4 positions before the hump (**b**) $x/c = -2$ (**c**) $x/c = -1.5$ (**d**) $x/c = -1.0$ (**e**) $x/c = -0.5$.

In Figure 6, it is evident that the results are converged when denser mesh is used. However, even though the mesh resolution is fine enough, the simulation results could not fit the theoretical data generated from the law of the wall. Lund method introduce artificial periodicity in the turbulence generation progress, the results are hardly improved by increasing the simulation time. However, the calculation results of mean velocity profiles gradually approach the law of wall at four positions along the streamwise direction, from $x/c = -2$ to $x/c = -0.5$.

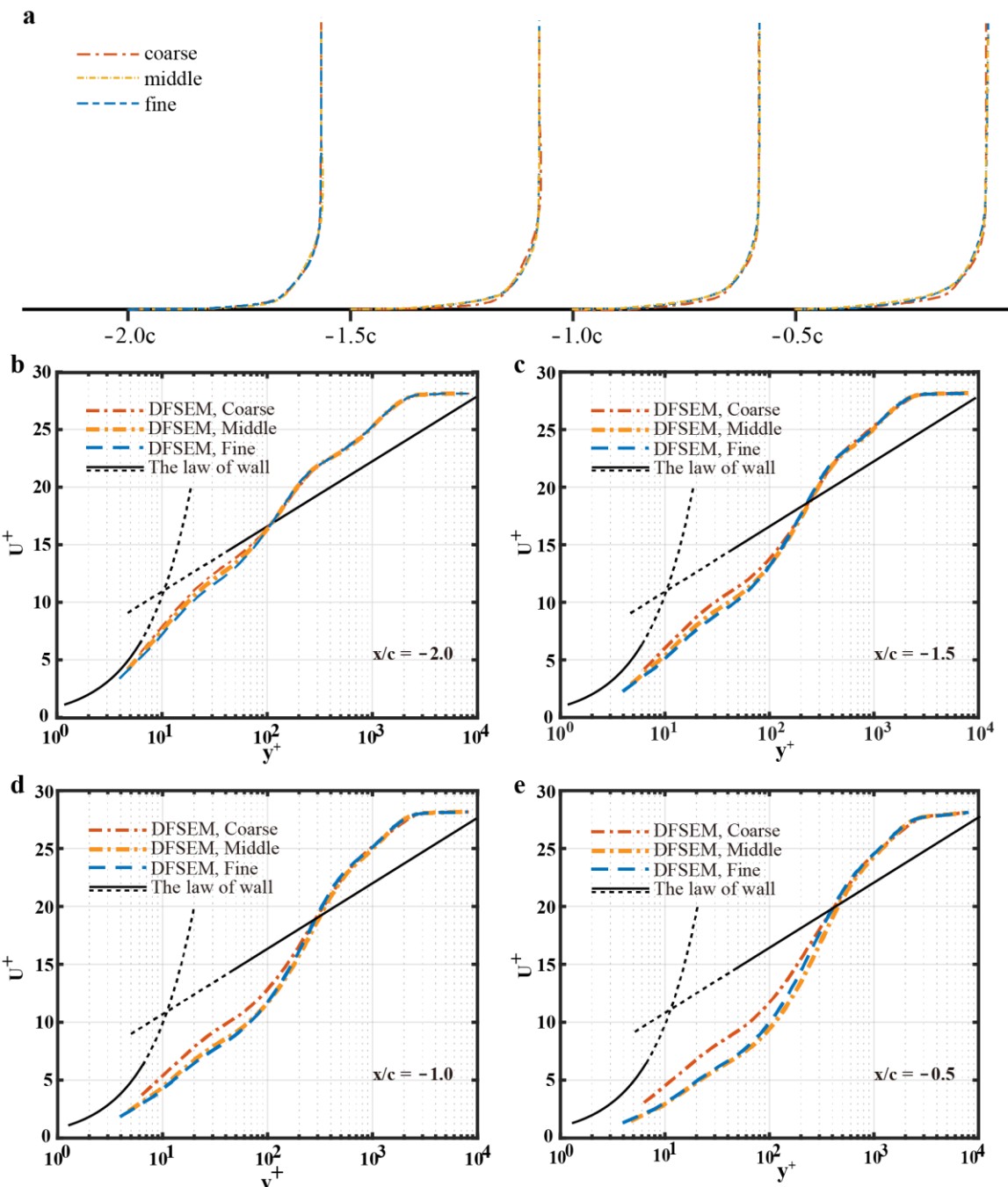

**Figure 7.** Effect of mesh resolution on streamwise mean velocity based on DFSEM at (**a**) 4 positions before the hump (**b**) $x/c = -2$ (**c**) $x/c = -1.5$ (**d**) $x/c = -1.0$ (**e**) $x/c = -0.5$.

In Figure 7, the calculation results of mean velocity profiles converge when denser mesh are used. However, the mean velocity profiles of the simulation become further away from the theoretical result as the flow moves from the inlet to the hump at four streamwise positions from $x/c = -2$ to $x/c = -0.5$.

In Figure 8, similarly, the calculation results converge as the mesh density increases. The distribution of mean velocity profiles at four streamwise positions has the same trend as the simulation based on DFSEM. Mesh sensitivity analysis after the hump is illustrated by Figures A1–A3 in the Appendix A.

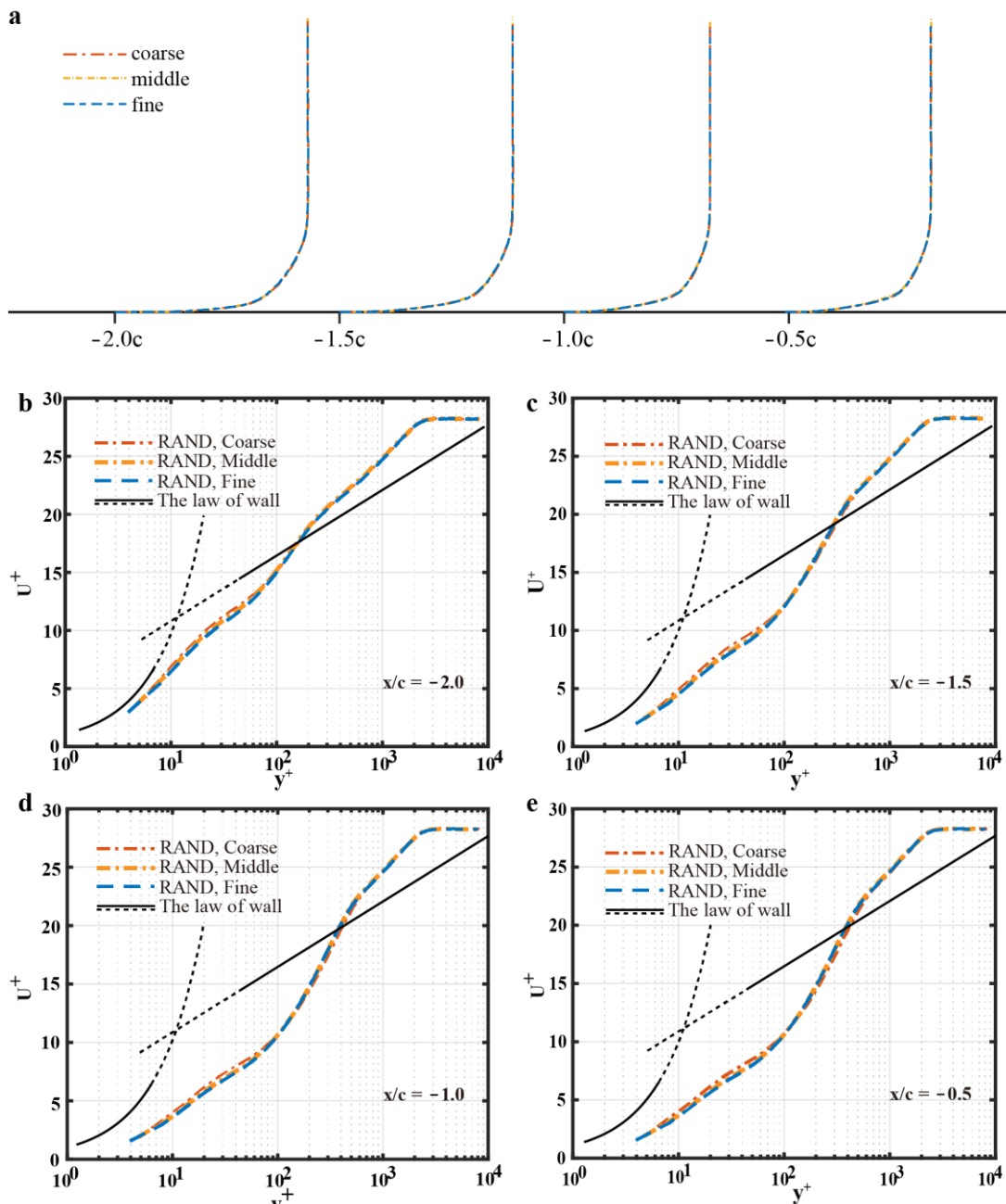

**Figure 8.** Effect of mesh resolution on streamwise mean velocity based on RAND method at (**a**) 4 positions before the hump (**b**) $x/c = -2$ (**c**) $x/c = -1.5$ (**d**) $x/c = -1.0$ (**e**) $x/c = -0.5$.

### 5.2. Global Velocity Field

Instantaneous streamwise velocity contours and the development of turbulent structure in the boundary layer from $x/c = -2.14$ (inlet plane) until $x/c = 4$ have been demonstrated for 3 IBCs in Figure 9 a–c. The instantaneous velocity field is captured at the time $T = 3T_A$, where $T_A$ is the flow-through time, $T_A = L/U$. The $Q$ criteria for three cases is the same and set as $Q = 10,000$. $Q$ is calculated from:

$$Q = \frac{1}{2}\left(\left\|\Omega\right\|^2 - \left\|S\right\|^2\right) \tag{20}$$

where $\Omega$ and $S$ are the antisymmetric and symmetric parts of the velocity gradient tensor [39]. The Q-criteria defines a vortex as a "connected fluid region with a positive second

invariant of $\nabla u''$. $Q$ represents the vorticity magnitude as greater than the magnitude of the strain rate, which can be used to indicate the turbulent structures in the flow region.

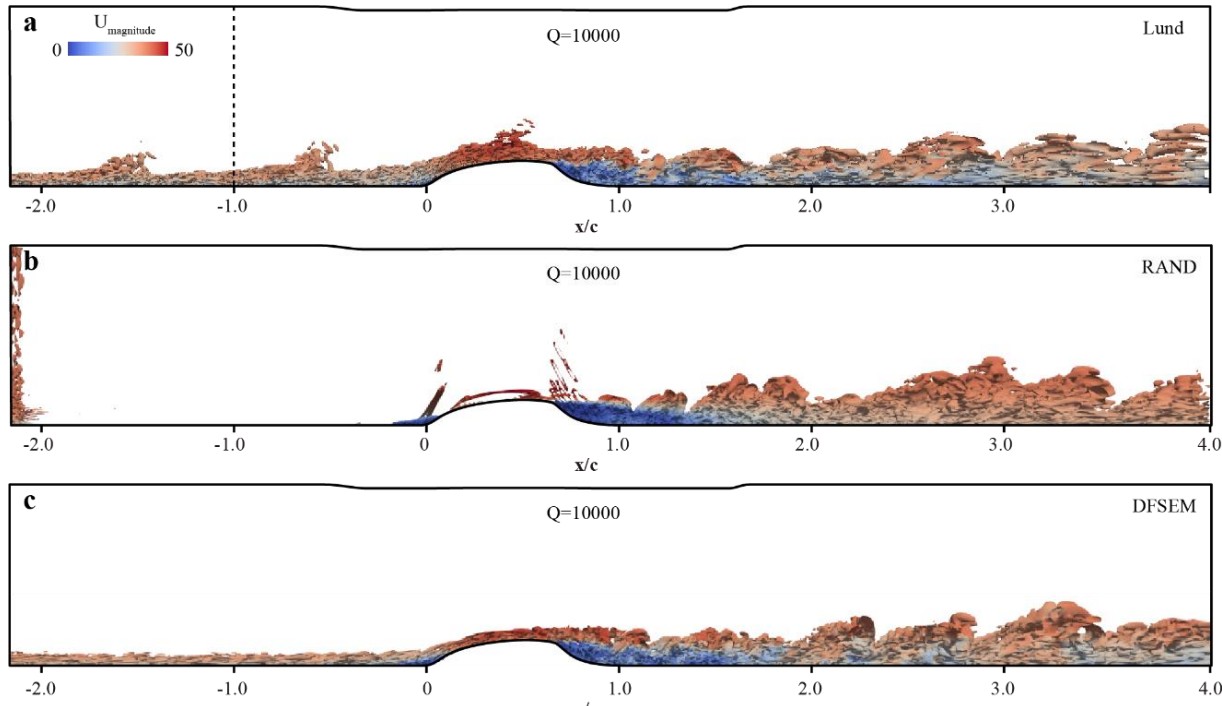

**Figure 9.** Instantaneous velocity field (**a**–**c**) at $T = 3T_A$ and the corresponding iso-surface at the inlet by using the (**a**) Lund, (**b**) RAND, and (**c**) DFSEM.

For the case of Lund, turbulence is generated in the recycling and rescaling process and then reintroduced in the flow region (Figure 9a). The dot line in Figure 9a illustrates the position of recycling plane. Turbulence are well developed at the beginning of the flow region and the regular distributed coherent eddies can be observed in the flow region. Moreover, turbulence is generated close to the walls. Turbulence can be maintained over the hump. However, there will be an artificial frequency introduced to the flow region as the boundary layer flow is repeatedly rescaled and fed back again and again. The effect of the artificial frequency is magnified as the distance between the inlet plane and the recycling plane decreases, but it never vanishes even though the distance is long enough. This problem was elucidated and nominated in [14] as 'spurious' periodicity. This periodicity interacts with some physical frequency of the flow, for instance, the frequency of vortex shedding instability. In addition, the distance between the inlet plane and the recycling plane is determined artificially, which brings uncertainty of simulation results. If this distance is short, there will be less space for turbulence to develop and reach stabilized integral quantities like momentum thickness, on the other hand, there will be more difficulty to obtain a converged solution in the limited distance. Otherwise, if the distance is very long, the computational cost tends to increase sharply. Regardless of whether the distance is short or not, artificial periodicity is inevitable.

For RAND method, some perturbations can be observed close to the inlet plane and in the middle of the vertical direction (Figure 9b). These disturbances are further weakened downstream and it is hardly to observe evident and stabilized turbulent structures before the hump ($x/c = -1 \sim -0.3$). The stabilized turbulent structures can be observed only after the hump (Figure 9b). Instantaneous velocity data are generated stochastically, which makes the energy of turbulence signal uniformly distributed over the whole wavenumber range. Due to a lack of energy in the low wave number range, the pseudo turbulence generated in this range damps to zero immediately. The results derived from this method are identical to a laminar flow.

For the case of DFSEM shown in Figure 9c, turbulence is developed from the anisotropic turbulent spots randomly distributed at the inlet plane. Turbulent structures are homogeneous before the hump and the thickness of the 'turbulent region' maintains the value of boundary layer thickness at the inlet. Compared with the Lund method, there is no periodic 'turbulent wave' when DFSEM is applied.

For all cases, 'turbulent region' is much thicker after the hump than that before the hump. It reflects that the flow separation could stimulate much stronger turbulence than the artificial inlet boundary condition. Streamline distribution for the three cases are demonstrated in Figure 10, where the orange line represents experimental data and the black line represents simulation results.

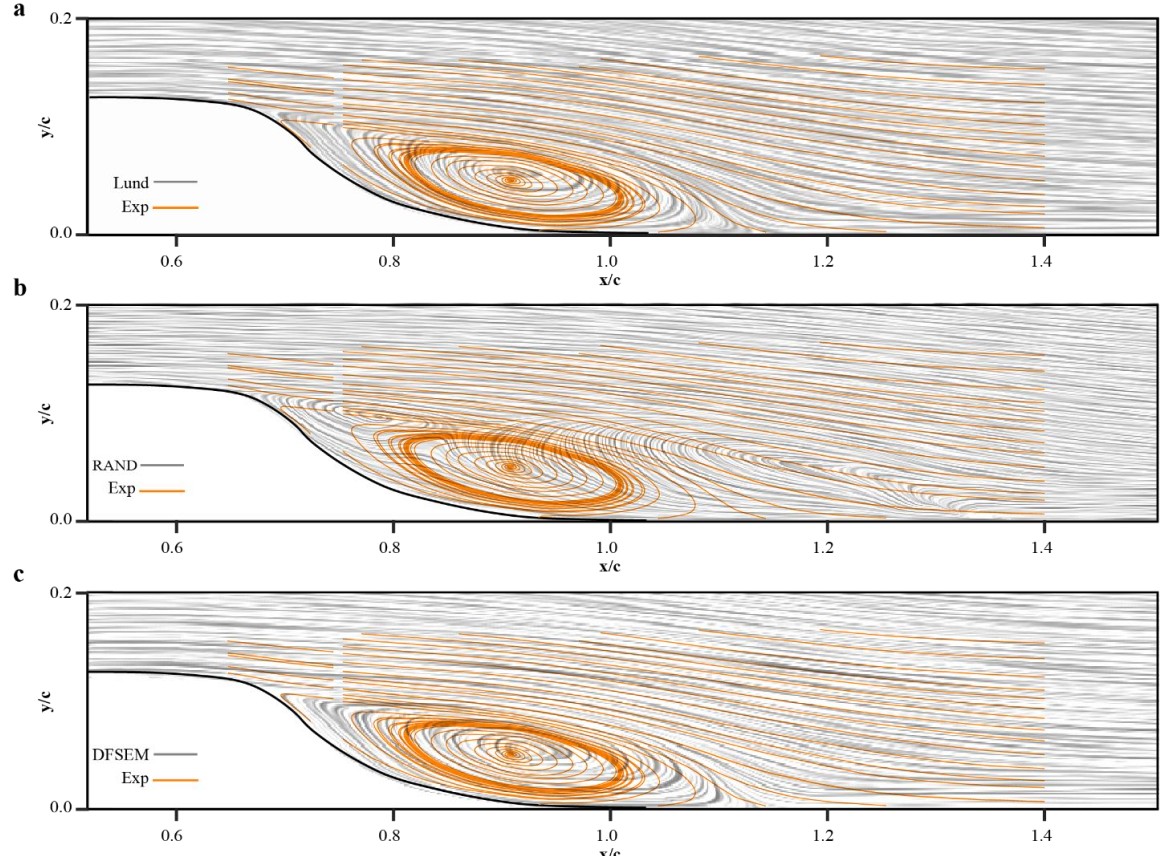

**Figure 10.** Streamline distribution around the lump plotted from (**a**) Lund method, (**b**) Random method and (**c**) DFSEM method.

It is apparent that the size of flow separation regions in the simulation based on the Lund method and DFSEM in Figure 10 are nearly the same. Besides, the streamlines of simulation based on the Lund method and DFSEM could fit the experimental data (Figure 10a,c) compared with the simulation based on the RAND method (Figure 10b). The position of the turbulent separation point of the case with the RAND method is close to others, while there are two vortices after the hump. Hence, the position of the reattachment point in this case is different from that in other cases.

### 5.3. Flow Separation and Reattachment (Pressure Coefficient and Friction Coefficient)

Table 3 presents flow separation and reattachment locations of three IBC methods. "Rel. diff to Exp" represents the relative difference between the simulation results and the experimental data. All these methods could give relatively accurate prediction of the positions of separation points. The relative difference between the prediction and the experiment data is smaller than one percent in cases based on the Lund method and DFSEM.

Simulations with all IBC methods overestimate the streamwise positions of reattachment points. In cases based on the Lund method DFSEM, the relative difference with respect to experiment data is nearly 6%. However, the value is huge in the case based on the RAND method, which is over 22%. Therefore, both simulations with Lund method and DFSEM could capture relatively accurate flow separation region, while the simulation with the RAND method greatly overestimated the size of the separation region after the hump.

**Table 3.** Separation and reattachment positions of three IBCs in the flow.

|  | Exp | Lund | DFSEM | RAND |
|---|---|---|---|---|
| Separation point ($x/c$) | 0.648 | 0.642 | 0.645 | 0.610 |
| Rel. diff to Exp (%) | — | 0.93 | 0.46 | 5.86 |
| Re-attach. Point ($x/c$) | 1.11 | 1.174 | 1.175 | 1.358 |
| Rel. diff to Exp (%) | — | 5.77 | 5.86 | 22.34 |

A comparison between simulated and measured values of pressure coefficient $C_p$ is illustrated in Figure 11, which demonstrates the time-averaged pressure coefficient distribution.

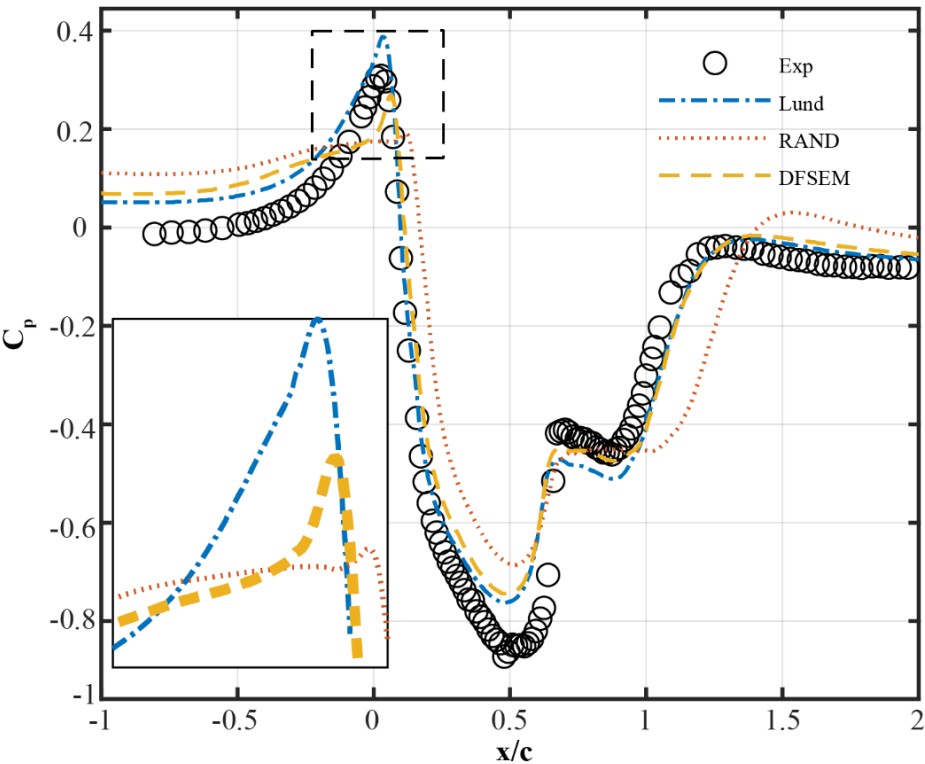

**Figure 11.** Distribution of pressure coefficient for three IBCs.

Before the hump, the pressure coefficient for Lund case is slightly larger than the experimental data and other simulations. The peak happened at the head of the hump in all three cases with different IBC methods, and the peak value of Lund case is slightly larger than the experimental value, on the contrary, the values of others are smaller than the experimental value. The information in the dotted line frame without experimental data is zoomed in and put in the bottom left of Figure 11. For the case based on DFSEM, it is obvious that the pressure coefficient yields a sharp increase at the position close to the head of hump and then it subjects to the same downward trend as the Lund case does after the peak and the pressure distribution is nearly the same. For the case based on the RAND method, slower growth could be observed before the head of hump compared with cases based on other IBC methods.

In cases based on the Lund method and DFSEM, the distribution of pressure coefficient shows a downward trend on the hump from $x/c = 0$ to $x/c = 0.5$ and then arrives at a 'valley' ($x/c = 0.5$). The values at this point in cases based on these two IBC methods are larger than experimental data. There is a second valley of the pressure coefficient after the hump. Meanwhile, the pressure distribution of cases based on the Lund method and DFSEM are close from $x/c = 0$ to $x/c = 2$. The distribution of pressure coefficient for simulations with RAND is different from simulations with Lund and DFSEM methods. The pressure coefficient is close to the experimental data before the position where $x = -0.5c$. The peak value obtained after the head of the hump, and the magnitude is much smaller than other simulation results and the experimental data. The valley located at $x = 0.5c$ same as other simulations and experiment, but the value is much larger. There is a second valley of the pressure coefficient after the hump, while the pressure coefficient after the second valley rises later than the experimental data.

Figure 12 demonstrates the time-averaged friction coefficient for different inflow methods. Before the head of the hump ($-1 \leq x/c \leq 0$), the friction coefficient keeps stable and then decreases for cases using the Lund method, while only the downward trend could be observed when applying RAND method and DFSEM as the inlet boundary condition method. When the flow reaches and cross the middle of the hump ($0 < x/c \leq 0.5$), the friction coefficient increases sharply in all cases at which $x/c \leq 0.15$ and reaches the peak, simulation based on the Lund method and DFSEM has the highest coefficient corresponding to a high rate of velocity acceleration of near-wall flow. The case using the RAND method shows a similar trend at $0 \leq x/c \leq 0.15$, while the peak value is smaller than that in the cases based on the Lund method and DFSEM. After the peak, the same downward trend of $C_f$ can be observed in cases using the Lund method and DFSEM. Simulation based on RAND method shows different rising and descending tendency at $0 \leq x/c \leq 0.15$ and $0.15 < x/c \leq 0.5$ compared to other cases. The growth rate at $0 \leq x/c \leq 0.15$ is smaller than simulations based on the Lund and DFSEM method, besides, the peak value is much smaller than other two cases. After the peak, at the region where flow separation shows up and flow crosses and leaves the second half of the hump ($x/c > 0.5$), $C_f$ shows a similar downward trend and the descending rate is nearly the same in all simulations based on different inlet boundary conditions.

The square (B, D), the circle (A, F), and diamond (C, E) marks in Figure 12 represent the flow separation and re-attachment point in simulations based on the Lund method, Rand method, and DFSEM respectively. In the case based on the RAND method, a flow separation point shows up firstly at the position A, where $x/c = 0.61$, and the value of friction coefficient decreases from positive to negative value, in other words, the near wall velocity gradient based on the distance to the wall has a sign reversal at this point. The friction coefficient keeps negative and stable and then drops rapidly until $x/c = 1.173$. The value of friction coefficient increases from negative value to positive value at the position F, where $x/c = 1.252$, which represents the flow re-attachment point. In the cases based on the Lund and DFSEM inlet boundary conditions, the positions of separation points at B, C and the positions of re-attachment points at D, E are very close. Therefore, the turbulent bubble sizes of these two simulations are close. This feature is illustrated in Figures 11 and 12, Table 3. The flow separation region in the simulation based on the RAND method is larger than that based on other IBC methods.

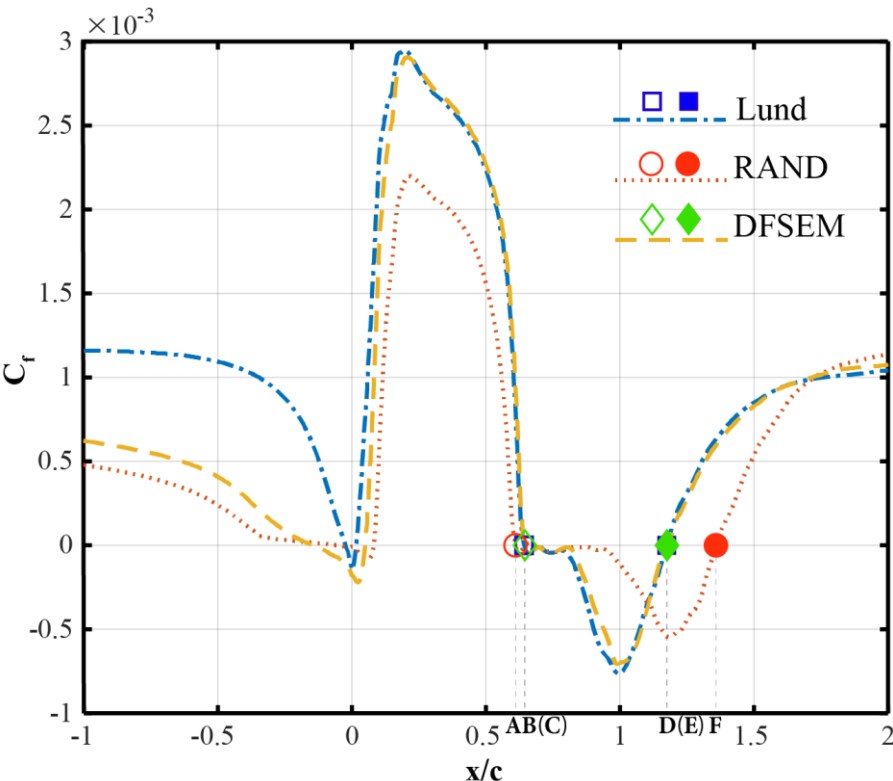

**Figure 12.** Distribution of friction coefficient for three IBCs (A = 0.610, B = 0.642, C = 0.645, D = 1.174, E = 1.175, F = 1.358).

Differences between the flow pattern after the hump for the simulation based on different IBC methods reflect that turbulence generated at the inlet could affect the flow separation after the hump. However, compared with the cases based on RAND method and DFSEM respectively, the distribution of pressure coefficient and drag coefficient shows the same trend before hump until *x/c* = 0. When flow passes through and leaves the hump, the distribution of pressure coefficient and drag coefficient are different in these two cases. The scenario is contrary when comparing the cases based on DFSEM and Lund method respectively. There is a large deviation between these two coefficients, $C_p$ and $C_f$ before the hump, while the values of these two coefficients are nearly the same over and after the hump.

*5.4. Quantifications of the Turbulent Properties before the Hump*

5.4.1. Time-Averaged Velocity Profiles before the Hump

From the perspective of time-averaged velocity distribution and statistical turbulence characteristics (such as Reynolds stress), it is very important to understand the behaviour of different IBCs and their influence on flow separation and re-attachment positions.

Figure 13 shows the time-averaged streamwise velocity profiles of different IBCs before the hump and compares them with theoretical results. Here the profiles in Figure 13a are plotted in uniform coordinate while the profiles in Figure 13b–e are plotted in logarithmic coordinate. For the simulation based on the Lund method, there is always a small deviation between the Lund results and the theoretical time-average velocity profiles. At *x/c* = −2, the deviation of the time-average velocity profile from the theoretical result is almost 0 at the viscous sublayer region ($y^+ \leq 5$) but large at the log-law region ($y^+ \geq 30$). At *x/c* = −1.5, the time-averaged velocity profile is consistent with the theoretical data in the viscous sublayer, meanwhile the velocity profile in the log-law region approaches the theoretical results. At *x/c* = −1 and *x/c* = −0.5, the time-averaged velocity profiles become much closer to the theoretical results though there is still a deviation between the simulation

result and the theoretical result. For the case based on RAND method, the time-averaged velocity profiles close to the wall at $x/c = -2$ has small deviation compared with other simulations and theoretical results, while the deviation becomes larger as the distance from the wall increases. At $x/c = -1.5$, the deviation of the meantime-averaged velocity profile from the theoretical result becomes larger. The same description can be applied to the time-averaged velocity profile at $x/c = -1$, and $x/c = -0.5$. The phenomenon is caused whereby the turbulence is not fully developed in the region before the hump. However, for DFSEM, the distribution of velocity profiles is close to that in the case based on the RAND method. This reflects that there are coherent structures in this region, but the turbulent intensity seems to be small, which makes the flow perform like laminar flow.

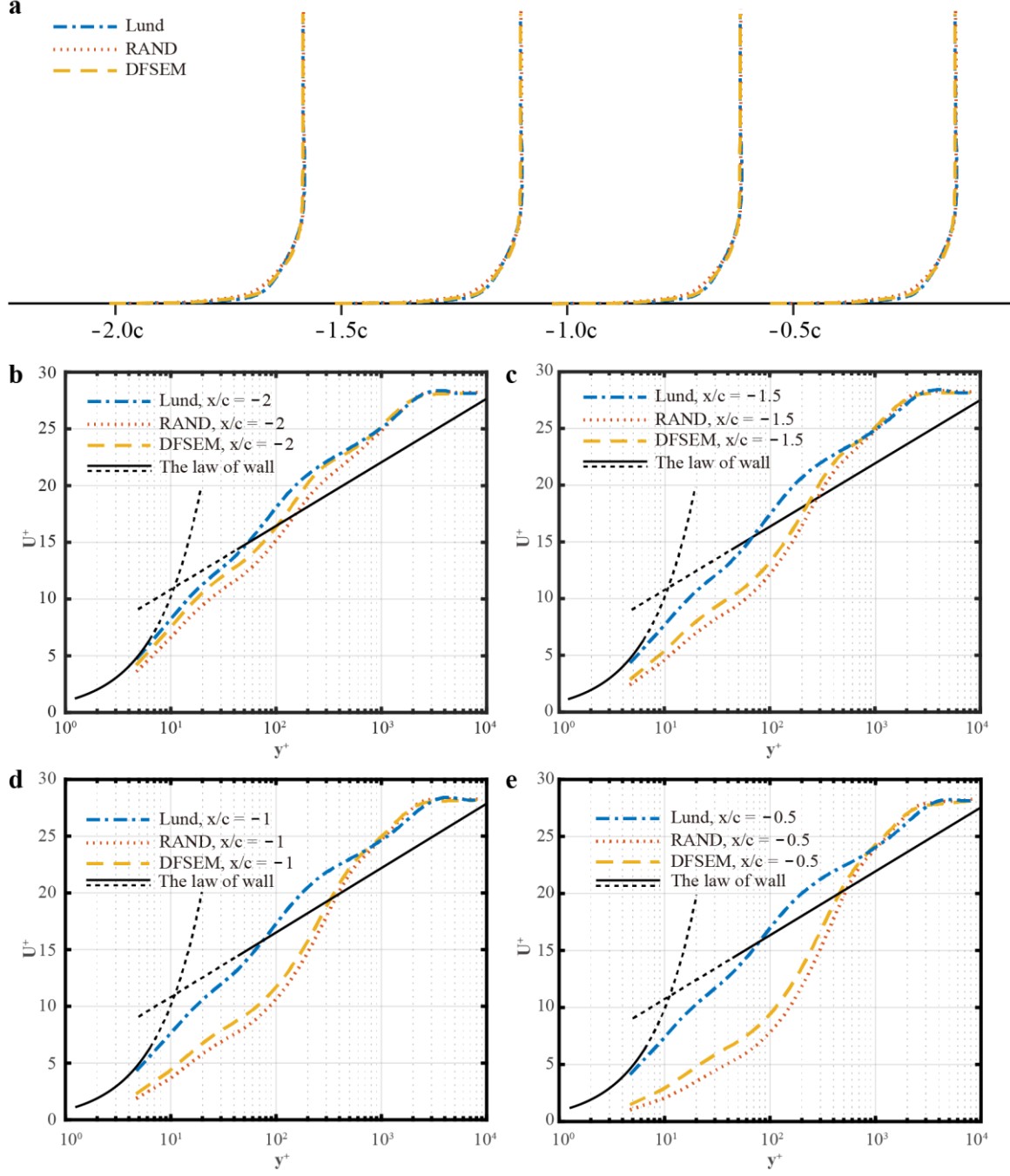

**Figure 13.** Time-averaged streamwise velocity U+ profiles (**a**) at four stations before the hump: (**b**) $x = -2c$, (**c**) $x = -1.5c$, (**d**) $x = -1c$, (**e**) $x = -0.5c$.

### 5.4.2. Reynolds Stress Profiles before the Hump

Figure 14 shows the Reynolds stresses profiles of simulation results based on different IBCs before the hump.

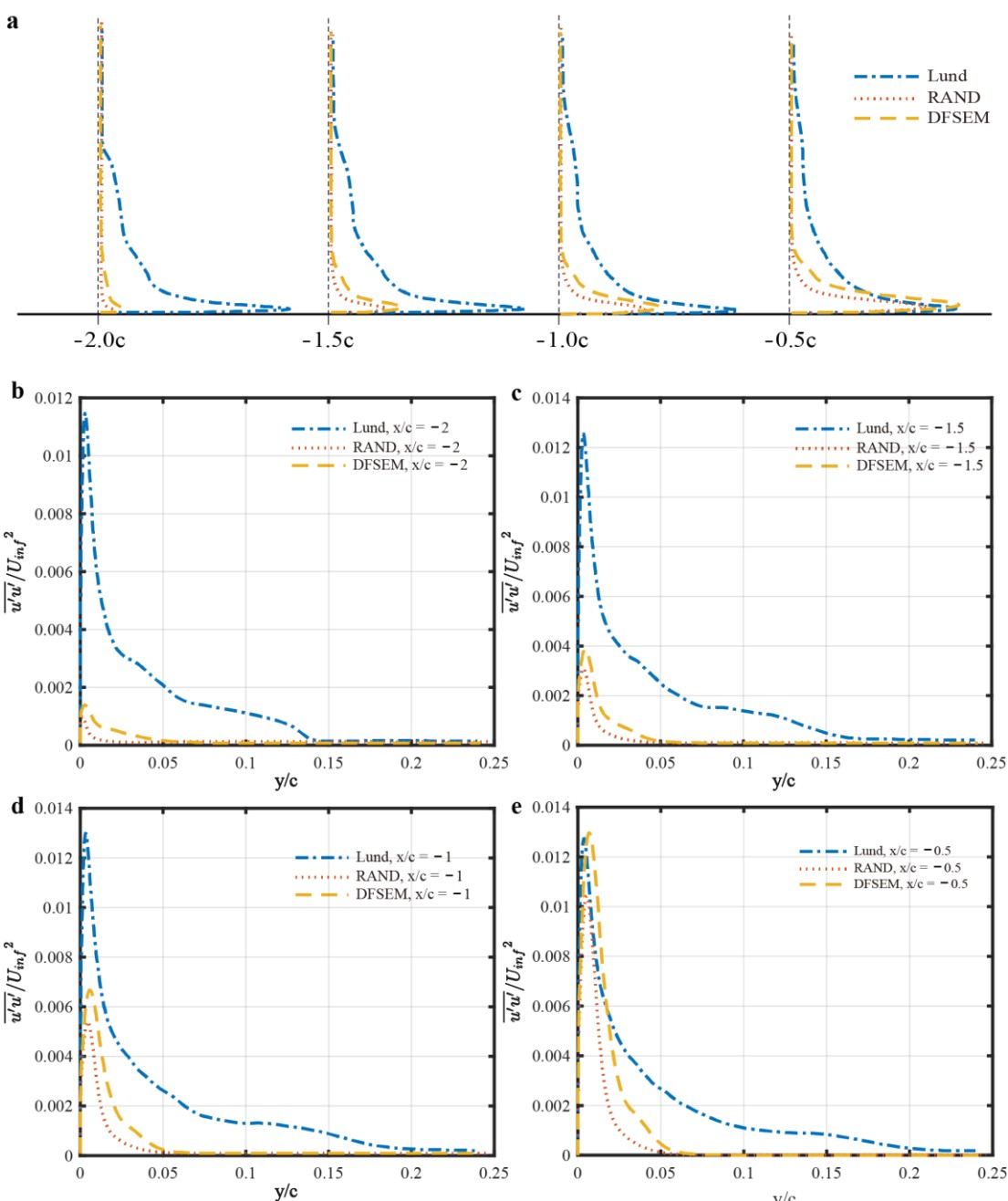

**Figure 14.** Time-averaged streamwise Reynolds stress $\overline{u'u'}/U_{inf}^2$ profiles at: (**a**) four stations before the hump (**b**) $x/c = -2$ (**c**) $x/c = -1.5$ (**d**) $x/c = -1$ (**e**) $x/c = -0.5$.

Since the values of $\overline{u'v'}$, $\overline{v'v'}$ and $\overline{w'w'}$ are considerably smaller compared with $\overline{u'u'}$ at the ahead of the hump, it is reasonable to use $\overline{u'u'}/U_{inf}^2$ to reflect the turbulent intensity and fluctuating part of the instantaneous velocity in the flow region. For the case based on Lund method, the profiles of $\overline{u'u'}/U_{inf}^2$ remain stable and slightly changes at different streamwise positions ($x/c = -2, -1.5, -1$ and $-0.5$). For the case based on RAND method and DFSEM, the streamwise turbulence intensity increases from $x/c = -2$ to $x/c = -0.5$, there is slight difference between the results of DFSEM and

RAND method. Figures 13 and 14 reflect that the profiles of mean velocity and streamwise turbulent intensity of the cases based on DFSEM and RAND method are similar before hump. According to Figure 9, the fluctuating velocity is not coherent in the case based on RAND method even though the turbulent streamwise intensity increases, while coherent structures can be observed in the case based on DFSEM.

*5.5. Quantifications of the Turbulent Properties over the Hump*

5.5.1. Time-Averaged Velocity Profiles over the Hump

In Figure 15a, the time-averaged velocity profiles over and after hump obtained by different IBCs cannot fully match the experimental data at four positions along the streamwise direction. In Figure 15b–e, even though the time-averaged velocity profiles and Reynolds stresses profiles are different before the hump, these profiles of cases based on the Lund method and DFSEM are close at certain streamwise positions over and after the hump. At $x/c = 0.65$, all the mean velocity profiles are lower than the experimental profiles in the near wall region but close to the experimental profiles when $y^+ > 500$. The deviation of simulation and experiment is much larger in the case based on RAND method. For all the cases at positions where $x/c = 0.8$ and 1, the mean velocity profiles are close to the experimental data in the near wall region but larger than experimental data when $y^+ \geq 4000$. Velocity profiles in cases based on the Lund method and DFSEM are almost coincidental, whereas that in case based on RAND method differs from the results of cases using the Lund method and DFSEM. Meanwhile, the differences between the results of the cases using the RAND method and the experimental data are larger than those in other cases. At the position $x/c = 1.2$, similarly, the simulation results of cases using Lund method and DFSEM are nearly the same and the results of the case using RAND method differs from the results of other cases. Considering the difference between the simulation and the experimental results, it is obvious to see that the case using the RAND method does not perform as good as cases with other methods.

5.5.2. Reynolds Stress Profiles over the Hump

By looking at the scales of the graph, streamwise stresses have significantly increased for all methods after flow separation compared with those values before hump. The region close to the surface ($y/c < 0.12$) has gained higher stresses compared to preceding station, which is due to approaching and impingement of separated shear-layer on the surface [40]. At the four sample positions ($x/c = 0.65$, 0.8, 1 and 1.2) shown in Figure 16, profiles of $\overline{u'u'}/U_{inf}^2$ show the same trend for simulations based on three IBC methods but the values are different. The main difference between the simulation results based on all IBC methods and the experimental data exists in the near wall region ($y/c < 0.12$). At the position $x/c = 0.65$, there is a slight drop of the peak values of $\overline{u'u'}/U_{inf}^2$ in cases based on the RAND method, Lund method, and DFSEM, respectively, and all peak values from simulations are larger than the experimental data. At the position $x/c = 0.8$, the distribution of cases based on all IBC methods are similar as well. Results of cases based on the Lund method and DFSEM are close, while the result of case based on RAND differs from simulation with other methods but becomes much close to the experimental data. At $x/c = 1$, 1.2, the difference between simulation results and the experimental data becomes much larger compared with the results at $x/c = 0.65$ and 0.8. The most likely cause of the difference is that simulations using these IBC methods could not predict the position of re-attachment point as precisely as the position of the separation point. Compared with the case based on the RAND method and DFSEM, the case using the Lund method performs more reliably at these two positions.

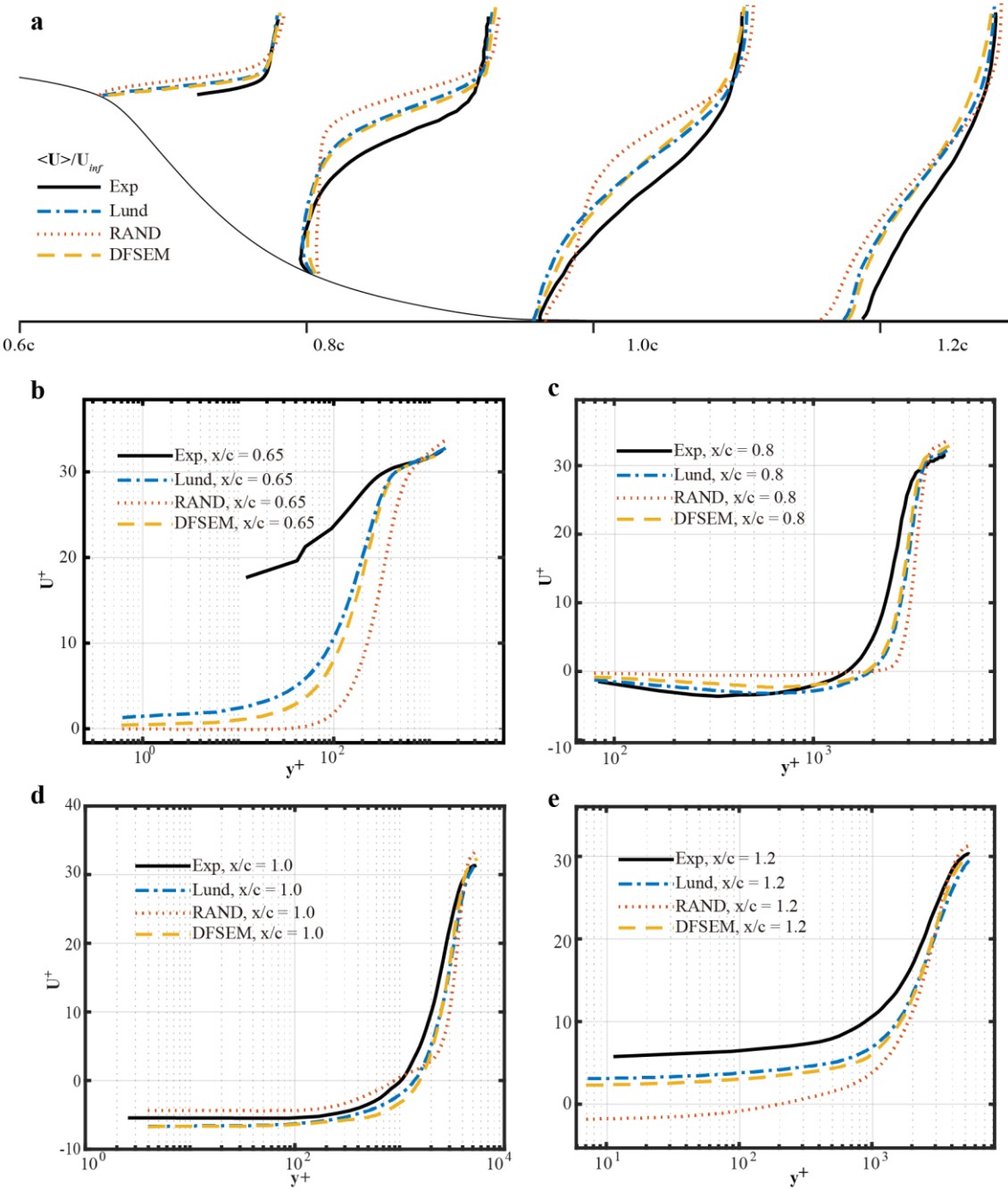

**Figure 15.** Time-averaged streamwise velocity $U^+$ profiles (**a**) at four stations over the hump (**b**) $x = 0.65c$ (**c**) $x = 0.8c$ (**d**) $x = 1.0c$ (**e**) $x = 1.2c$.

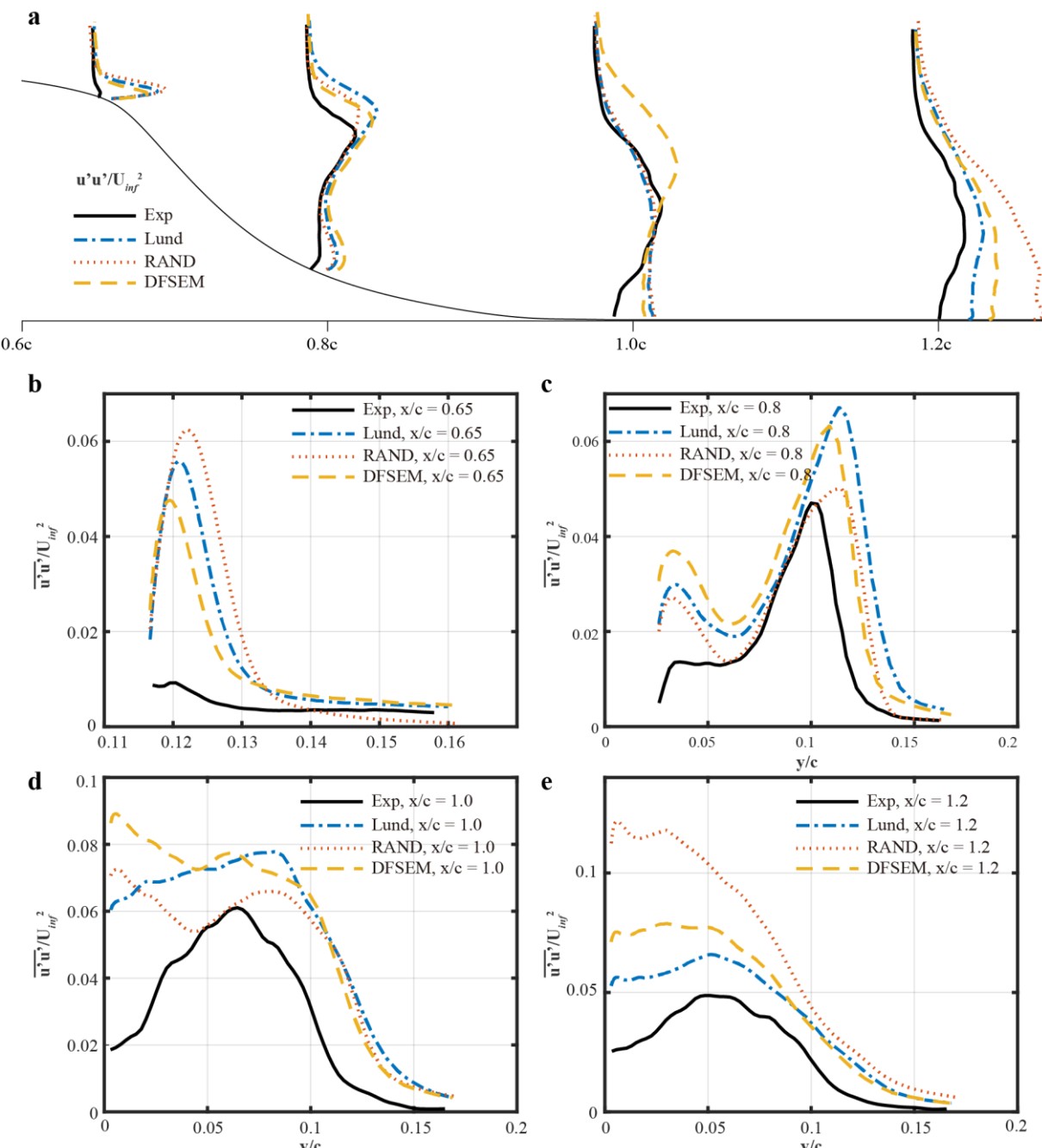

**Figure 16.** Time-averaged Reynolds stress component $\overline{u'u'}/U_{inf}^2$ profiles (**a**) at four stations over the hump (**b**) $x = 0.65c$ (**c**) $x = 0.8c$ (**d**) $x = 1.0c$ (**e**) $x = 1.2c$.

Profiles of $\overline{v'v'}/U_{inf}^2$ for simulations with different IBCs and experiment are shown in Figure 17. At $x/c = 0.65$, the trend of the cases based on the Lund method and DFSEM are same to the experimental data, but the magnitude of $\overline{v'v'}/U_{inf}^2$ in cases based on these two methods is as twice as the experimental data when $y/c = 0.12\sim0.16$, the trend of the simulation based on the RAND method is different. At $x/c = 0.8$, there is a slight deviation between the cases based on the Lund method and DFSEM, the difference between the results of cases using three IBC methods are close to the experimental data. At $x/c = 1$ and 1.2, profiles of simulation case based on RAND method shows large difference to the experimental data, while the results of the simulation cases based on the Lund and RAND methods are closer to the experimental data compared with the simulation based on the RAND method.

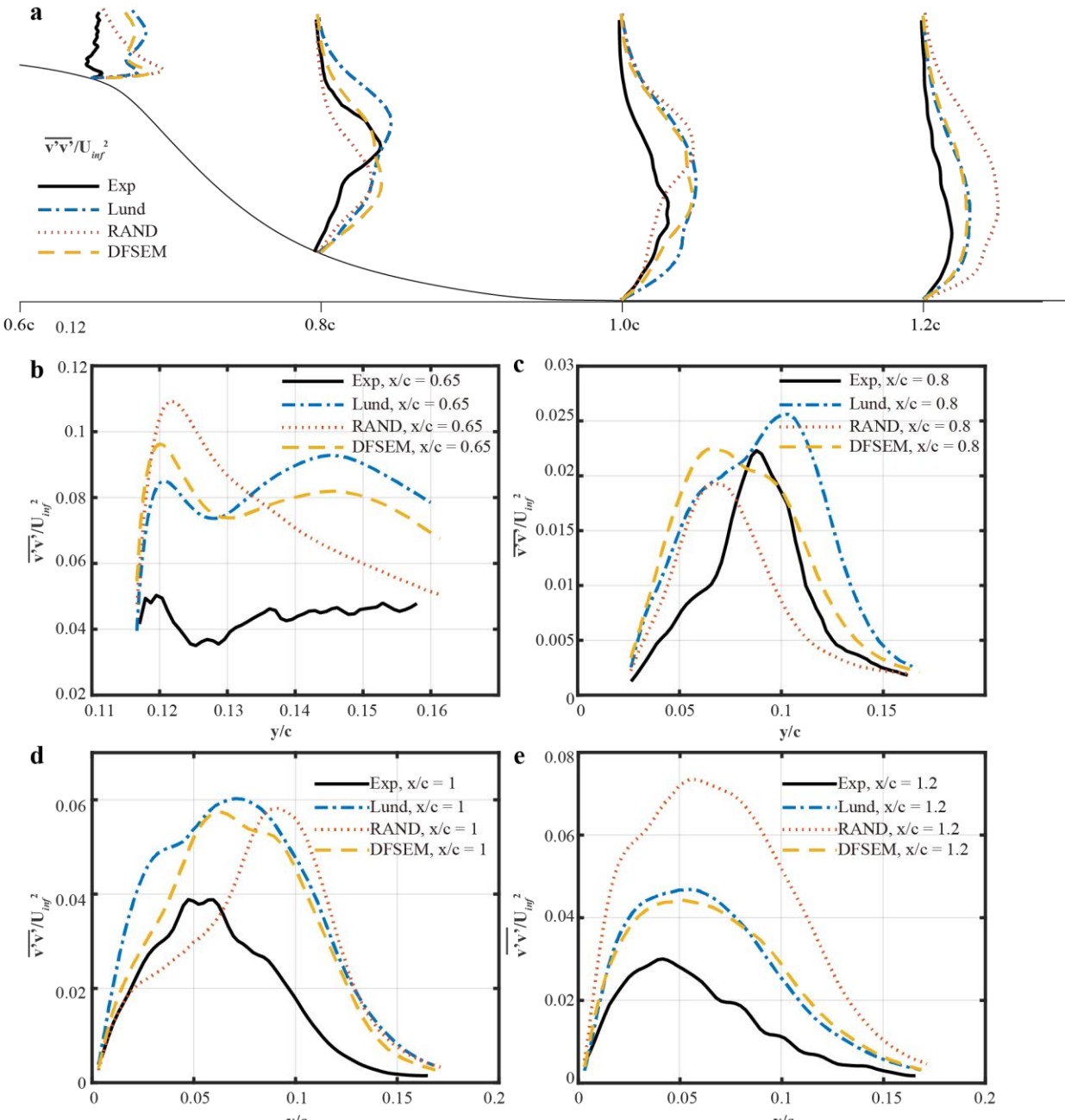

**Figure 17.** Time-averaged Reynolds stress component $\overline{v'v'}/U_{inf}^2$ profiles (**a**) at four stations over the hump (**b**) $x = 0.65c$ (**c**) $x = 0.8c$ (**d**) $x = 1.0c$ (**e**) $x = 1.2c$.

The distribution of $\overline{u'v'}/U_{inf}^2$ along the vertical direction illustrated by Figure 18 shows the similar behaviour as the profiles of $\overline{u'u'}/U_{inf}^2$ and $\overline{v'v'}/U_{inf}^2$, at the flow separation region ($x/c = 0.65$ and 0.8). The results of simulation based on the Lund method and DFSEM are closer to the experimental data than the results of simulation based on the Lund and RAND methods, but the deviation is still large at $x/c = 0.65$. At $x/c = 0.8$, the simulation results based on three IBC methods are close to the experimental data, especially the case based on DFSEM. In the flow re-attachment region ($x/c = 1$ and 1.2), the simulation results based on three IBC methods are closer compared with the results in the flow separation region. There is always a slight deviation between the simulation results and the experimental data.

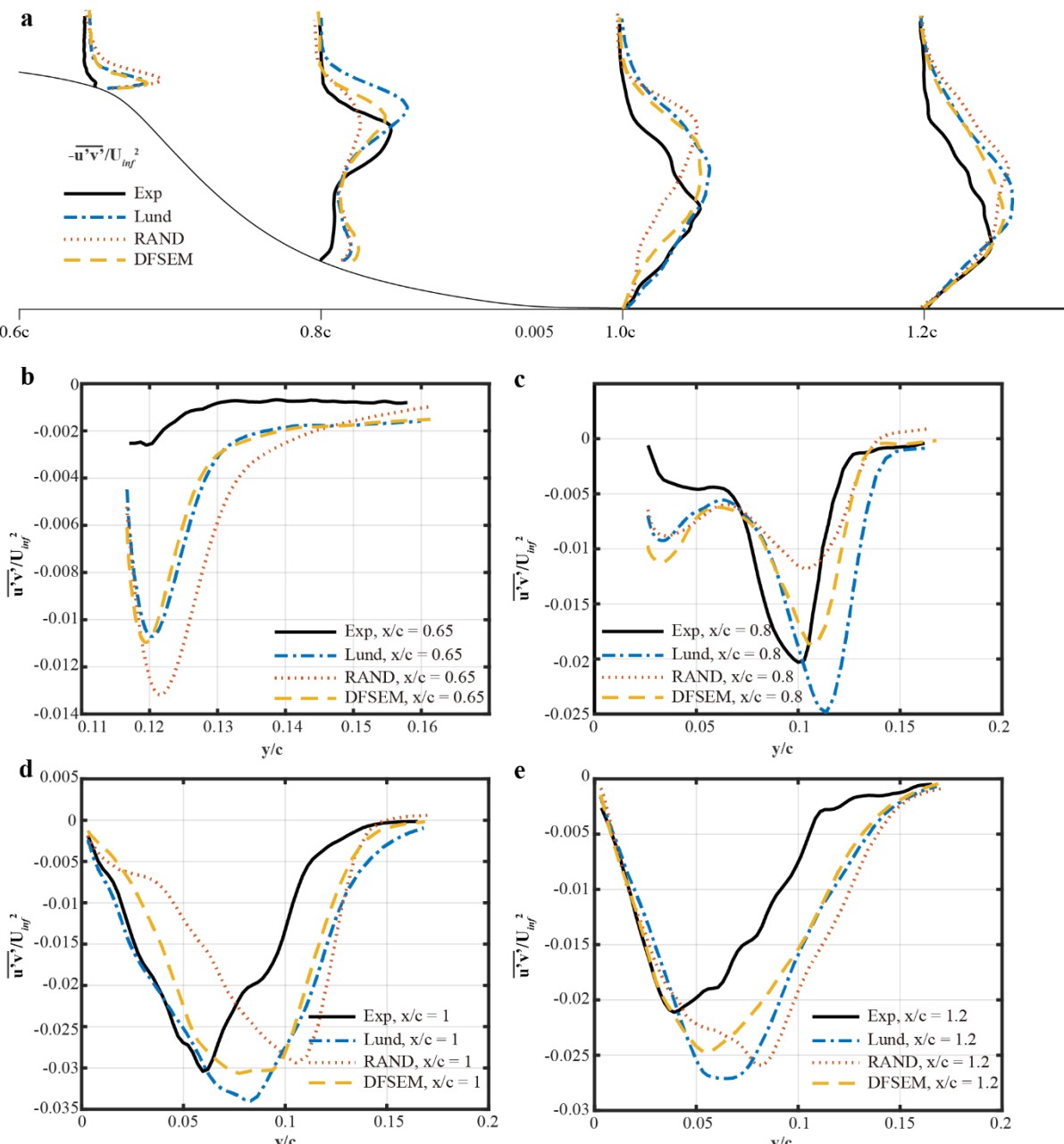

**Figure 18.** Time-averaged Reynolds stress component $\overline{u'v'}/U_{inf}^2$ profiles (**a**) at four stations over the hump (**b**) $x = 0.65c$ (**c**) $x = 0.8c$ (**d**) $x = 1.0c$ (**e**) $x = 1.2c$.

## 6. Conclusions and Further Work

In this study, the impact of three different IBC methods used in LES in the boundary-layer flow with a hump were investigated. $Re_\theta$ at the inlet in this flow configuration is 6473.5. It is challengeable to capture the flow details with the extensive flow separation region with this high Reynolds number. Nevertheless, the flow separation after the hump could reflect more details of the effects from different IBC methods.

From the qualitative results, turbulence generated from different IBCs at the region before hump shows apparent differences. In the simulation based on the Lund method, clear turbulent structure can be observed. Due to the artificial periodicity imposed in the flow region during this recycling and rescaling process, the turbulent signal is periodic. In the simulation based on RAND method, it is hard to see the clear coherent turbulent structure before the hump. As the flow develops and gradually approaches the hump, the

turbulence generated at the inlet quickly dissipates. In the simulation based on DFSEM, compared with the simulation based on the Lund method, coherent turbulent structures can be observed apparently in the boundary layer. In the region over and after the hump, the differences between the simulation results based on the Lund method and DFSEM method are not great as shown in Figures 9 and 10. However, there are still obvious differences between the simulation results based on the RAND method and simulation results based on other methods, which indicates that the coherent turbulence in the approach flow region will significantly affect the flow separation after the hump.

From the quantitative results, mean velocity and turbulent Reynolds stresses profiles are used to reflect the flow details of each simulation. In the simulation based on the Lund method, there is a deviation between the simulation and theoretical results at the beginning, while the deviation gradually decreases as flow moves towards the hump from $x/c = -2 \ to \ x/c = -0.5$. The scenario is different in the cases based on the RAND method and DFSEM. There is a slight deviation between the simulation results and theoretical results at $x/c = -2$, but the deviation increases as the main flow approaches the hump from $x/c = -2$ to $x/c = -0.5$. The statement above reflects that turbulence generated by Lund method is likely fully developed in the approach flow region. According to the differences of Reynolds stresses profiles in these simulations with different IBCs, it is obvious that the profiles of streamwise turbulent stresses in Lund method are maintained at different positions before the hump, while the significant changes can be found in the case of DFSEM, there is barely turbulence in the case of RAND at different positions before the hump. This phenomenon indicates that the generated turbulence is stable in the case of Lund, pseudo-developed in the case of DFSEM and not developed in the case of RAND. In the flow region after the hump, even if the location of flow separation is nearly the same in these three cases, the effects of different IBCs are more manifested. Firstly, there are not remarkable differences between the flow details of the simulations based on Lund and DFSEM methods. The distribution of the mean velocity and Reynolds stresses profiles are close in the simulations based on these two methods. The turbulent bubble size can be predicted relative precisely in these two simulations, there is only a slight deviation in the turbulent statistics profiles between the simulation results and the experimental data. Compared with other simulations, the simulation based on the RAND method shows obvious difference. It cannot capture the flow separation and re-attachment point. Meanwhile, turbulent bubble size in the base based on the RAND method is highly overestimated so that it cannot predict the velocity and Reynolds stresses profiles accurately.

From the analysis above, it can be found that the statistical properties of the case based on DFSEM shows a special trend before and after the hump as illustrated from Figures 11–15. In the approach flow, the mean velocity profiles of the case based on DFSEM cannot fit the theoretical results and performs like the simulation using the RAND method though the coherent structures could be observed in the approach flow. However, in the flow region over and after the hump, the results of the simulation using DFSEM become much close to the simulation results based on the Lund method, though the mean velocity profiles before the hump is significantly different between these two cases. These features indicate that the stable coherent structure before the hump is not fully developed turbulence, since the turbulent intensity is not large enough to make the mean velocity profiles fit the wall function. Nevertheless, the not fully developed turbulence is still able to delay the flow separation. Therefore, different turbulent inflow generation methods have a manifested impact on the flow separation after the hump. If the coherent turbulence is maintained in the approach flow, even though turbulent intensity is not large enough, the simulation can still predict the flow separation and turbulent bubble size relative precisely.

However, there are still some limitations of this study. Firstly, the spanwise width of the simulation domain is relatively small. In the future, spanwise width sensitivity analysis should be applied and a wider domain would be adopted. Secondly, the LES sub-grid scale model has a great impact on the results. In the future, a more advanced SGS

model such as the dynamic Smagorinsky model could be applied. Finally, in the simulation based on DFSEM, the setting of turbulent length scale at the inlet is an important factor affecting the simulation results. In the future, the influence of turbulent length scales should be considered.

**Author Contributions:** Conceptualization, Y.W. and H.H.; methodology, Y.W., G.V., B.F.; software, G.V., Y.W.; validation, Y.W., H.H., J.W.; formal analysis, Y.W., G.V.; investigation, Y.W., H.H., B.F.; resources, J.W., H.H.; data curation, Y.W.; writing—original draft preparation, Y.W.; writing—review and editing, H.H., B.F., G.V.; visualization, Y.W., G.V.; supervision, H.H., B.F., J.W.; project administration, Y.W., H.H.; funding acquisition, H.H., J.W. All authors have read and agreed to the published version of the manuscript.

**Funding:** This research received no external funding.

**Acknowledgments:** The authors are thankful to computational resources from BlueBEAR in the University of Birmingham (UoB) and "Taiyi" in Southern University of Science and Technology (SUStech), the work is sponsored by the partnership between the UoB and the SUStech.

**Conflicts of Interest:** The authors declare no conflict of interest.

## Nomenclature

| | |
|---|---|
| $P$ | pressure |
| $u$ | velocity |
| $\overline{u}$ | filtered velocity |
| $\tau_w$ | wall shear stress |
| $u_\tau$ | friction velocity defined by $u_\tau = \sqrt{\tau_w/\rho}$ |
| $\tau_{ij}$ | stress tensor |
| $\overline{S}_{ij}$ | filtered rate-of-strain tensor |
| $\nu_T$ | turbulent viscosity |
| $C_s$ | Smagorinsky coefficient |
| $\Delta$ | sub-grid length |
| $f_\mu$ | damping function |
| $y^+$ | non-dimensional near wall distance defined by $y^+ = u_\tau y/\nu$ |
| $k_{sgs}$ | the production rate |
| U | mean velocity |
| $U^\infty$ | free stream velocity |
| $\langle\rangle_z$ | average in the spanwise direction |
| $\gamma$ | the ratio of friction velocity at the inlet plane to the friction velocity at the recycle plane |
| $\delta$ | boundary layer thickness |
| $\delta^*$ | Displacement thickness |
| $\eta$ | non-dimensional wall distance defined by $\eta = y/\delta$ |
| $W(\eta)$ | weighting function |
| $\theta$ | momentum thickness |
| $Re_\theta$ | Reynolds number based on momentum thickness |
| $\Delta t$ | computational time step |
| $T$ | characteristic time scale of the averaging interval |
| $\alpha$ | weighting average factor |
| $R_{ij}$ | Reynolds stress tensor |
| $f_\sigma(x)$ | suitable shape function |
| $\varepsilon_j$ | random numbers with zero average |
| L | length of the channel with a hump |
| **Subscripts** | |
| *inlt* | at the inlet plane in Lund method |
| *recy* | at the recycle plane in Lund method |
| $\sigma$ | turbulent length scale in DFSEM |
| **Superscripts** | |
| *inner* | inner region in the boundary layer |
| *outer* | outer region in the boundary layer |
| ' | root mean square of fluctuating value |

## Appendix A

The grid independency analysis after the hump is shown in Figures A1–A3.

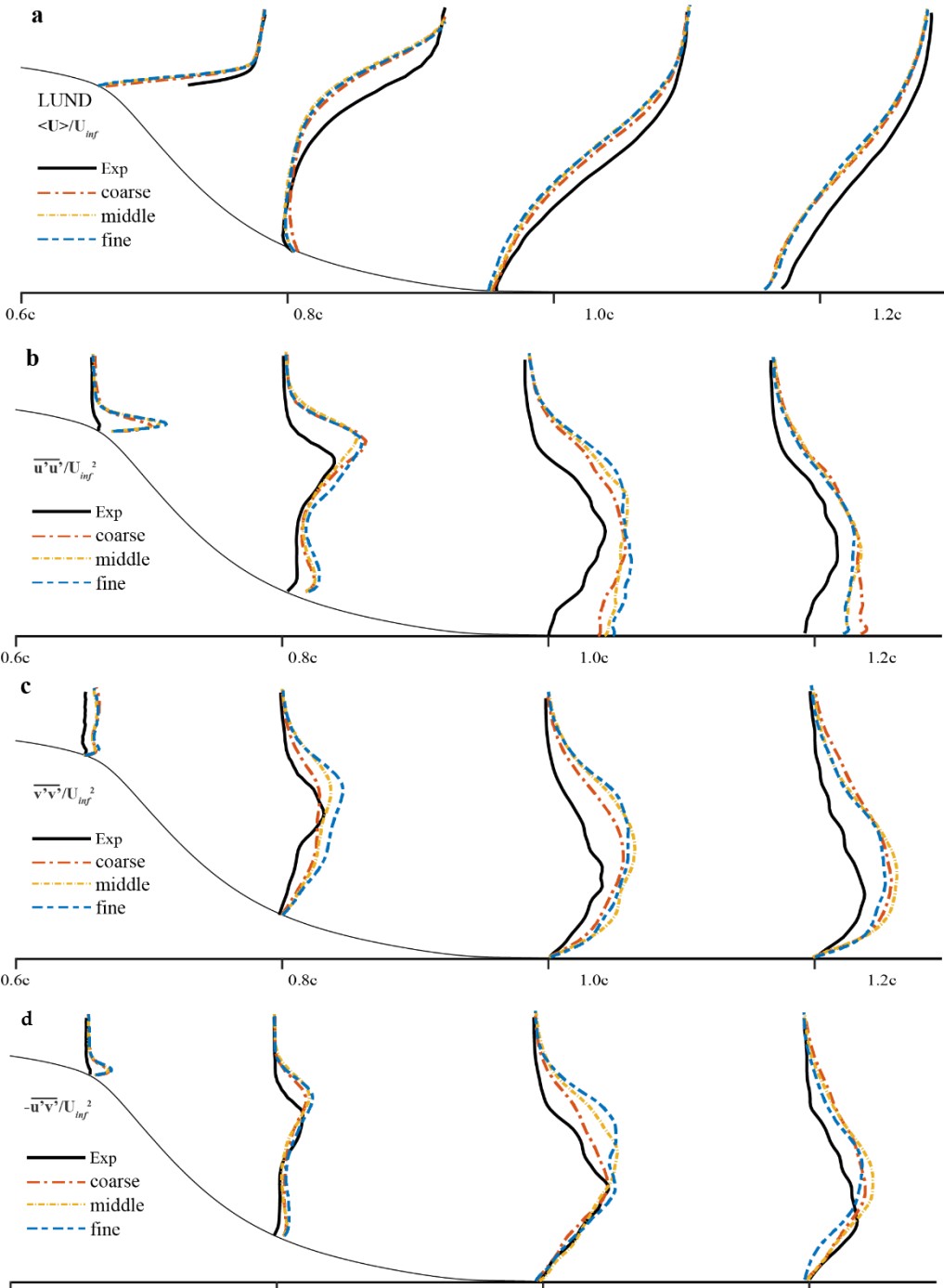

**Figure A1.** Grid independency analysis after the hump for the simulation based on Lund method: (**a**) Mean velocity profiles (**b**) profiles of Reynolds stresses component $\overline{u'u'}$ (**c**) profiles of Reynolds stresses component $\overline{v'v'}$ (**d**) profiles of Reynolds stresses component $\overline{u'v'}$.

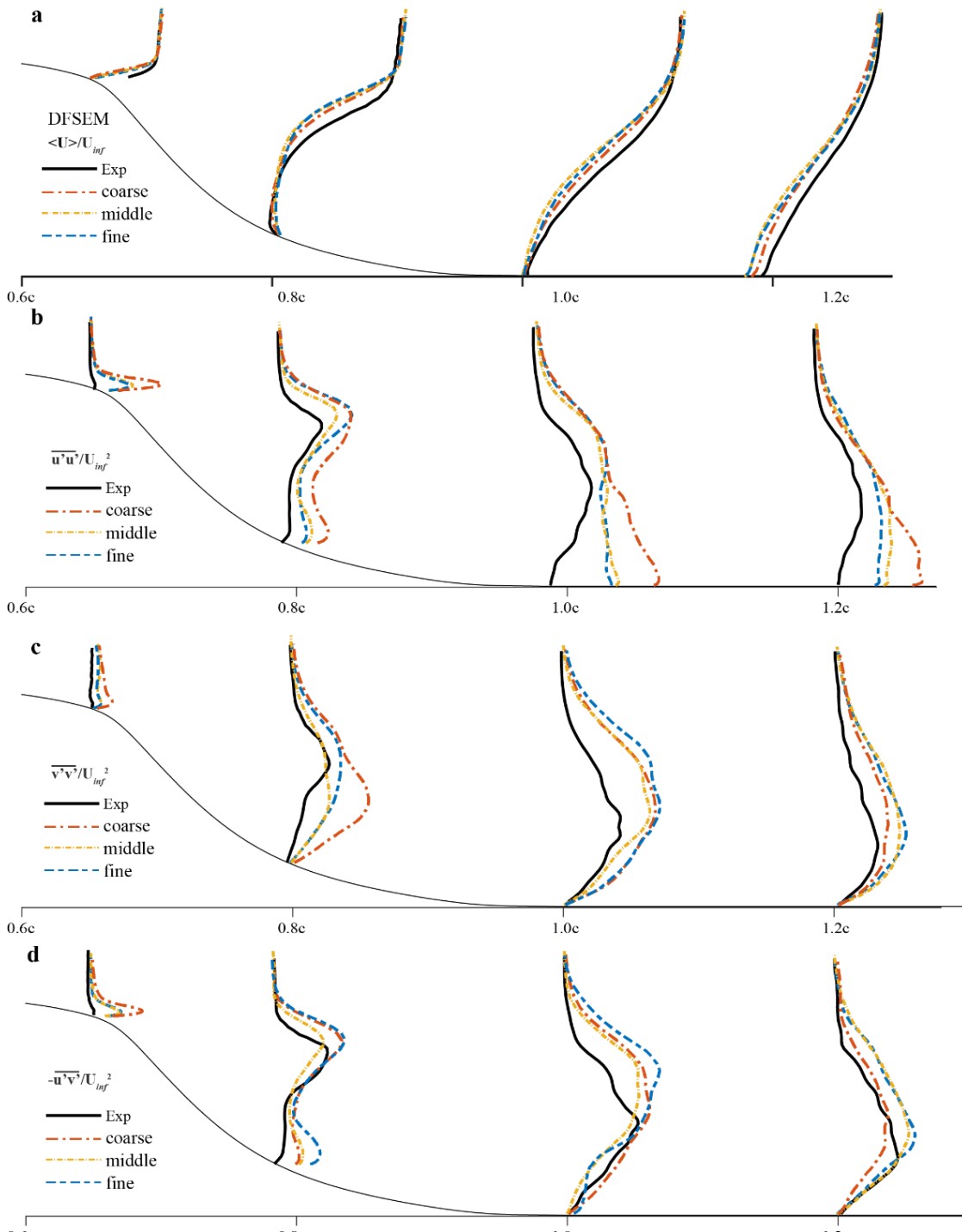

**Figure A2.** Grid independency analysis after the hump for the simulation based on DFSEM: (**a**) Mean velocity profiles (**b**) profiles of Reynolds stresses component $\overline{u'u'}$ (**c**) profiles of Reynolds stresses component $\overline{v'v'}$ (**d**) profiles of Reynolds stresses component $\overline{u'v'}$.

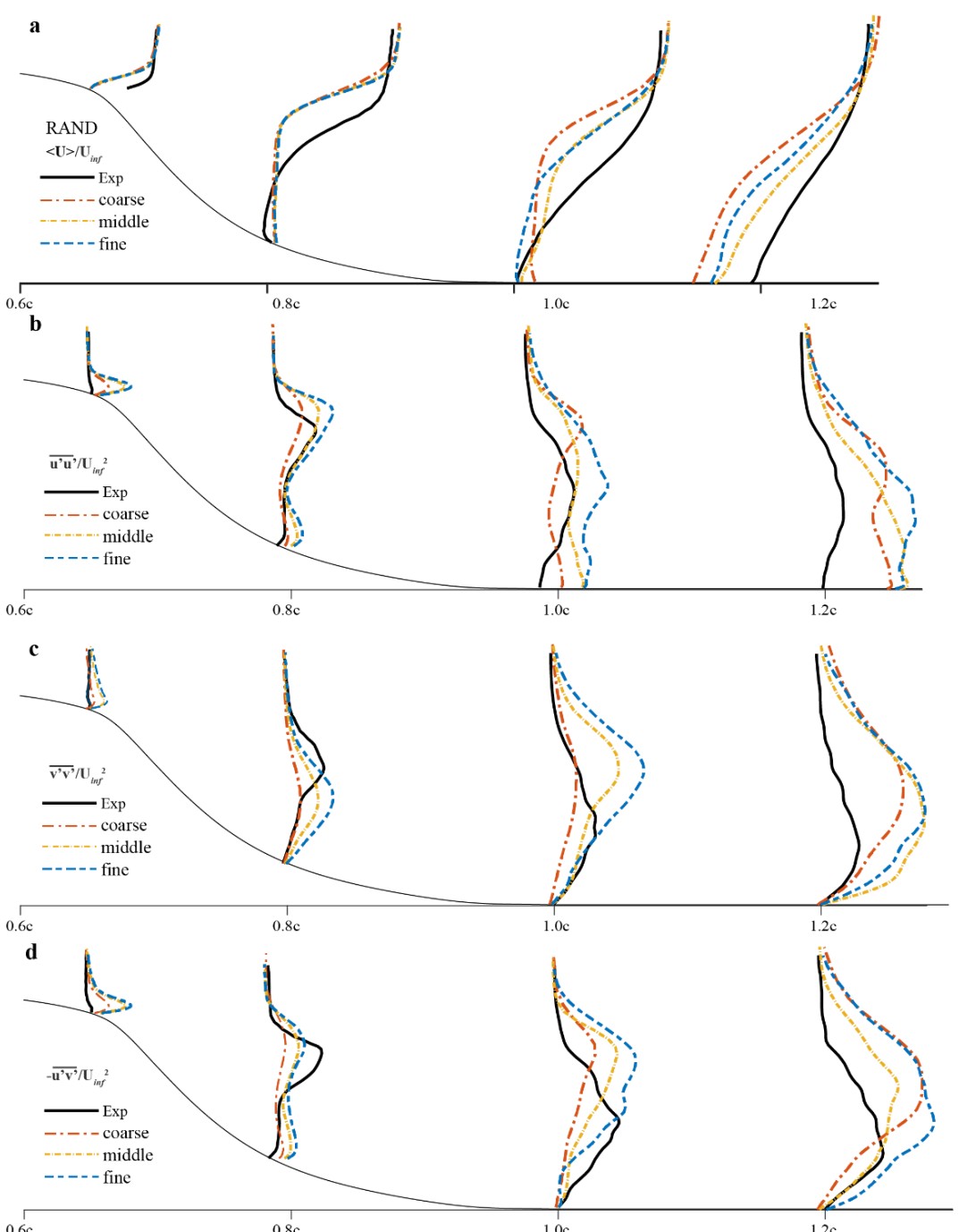

**Figure A3.** Grid independency analysis after the hump for the simulation based on RAND method: (**a**) Mean velocity profiles (**b**) profiles of Reynolds stresses component $\overline{u'u'}$ (**c**) profiles of Reynolds stresses component $\overline{v'v'}$ (**d**) profiles of Reynolds stresses component $\overline{u'v'}$.

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
