# Peer review of "Effect of the Inlet Boundary Conditions on the Flow over Complex Terrain Using Large Eddy Simulation"

_designs, 2020_

Round 1
Reviewer 1 Report
The manuscript focuses on the effect of different inlet boundary conditions in Large Eddy Simulations. The work is interesting and original. However, the following points need to be addressed prior to the publication of the work.
- Literature is limited. Please consider referencing and discussing relative works e.g. (Yang et al. 2020)(Vasaturo et al. 2018) (Tolias et al. 2018)(Aboshosha et al. 2015)(Wu 2017)(Bazdidi-Tehrani et al. 2016)(Yan et al. 2015).
- Section 3.2: Proper values of the coefficients in this formula can give coherent fluctuations and has been used successfully in (Tolias et al. 2018).
- How did you choose the dimensions of the domain? The length before the hump can be crucial for turbulence to develop and may impact the results. Have you conducted and domain-size sensitivity study? Why the Z dimension of the domain is so small? Turbulence can be dumped because it is like solving a 2D-problem.
- Why do you use Smagorinsky constant equal to 0.2? You should support your choice from the literature.
- Why you did not use a more advanced LES method, like the dynamic LES? Don’t you think that the results would be different? For example, the agreement with the experiment in Figure 9 could be changed.
- About temporal discretization, you mention that you use a backward differences scheme. What is the order of the scheme? A second order scheme is required for LES.
- Can you give some more details about the convection terms scheme? Is it a central differences one?
- Please provide some additional details about the mesh. Is the grid equidistant? Pictures of the grid (at the inlet area and around the hump) would be useful.
- The parameters of each IBC method that were used should be defined in the Section 4.2, e.g. what values of Reynolds stresses are used in DFSEM method. Please present also the experimental streamwise velocity fluctuations that you utilized.
- How did you obtain a consistent volume flow rate equal to 1.1 m3/s having different inlet velocity profiles?
- About grid independency analysis: Don’t you think that the used grids have similar number of cells and this is why small differences in the results are observed?
- Section 5.1: The authors need to explain why the simulation results could not fit the theoretical data generated from the law of the wall. Is it possible that the length of the domain is not enough (see also my previous relative comment)?
- Section 5.1: The results of the Random method are missing.
- Section 5.1: Why do you evaluate grid independency only on the area before the hump? Velocities and stresses above and after the hump, where experimental data exists, should also be evaluated in terms of grid independency.
- Figure 8: In Lund results, it seems that there is a discontinuity at x/c=2. Why is that happening?
- Figure 9: How do the predicted streamline distribution change when the denser grid was used?
- Figure 12a: It would be great if could add the experimental profile in the figure.
- Figure 13: Aren’t any experimental measurements available?
- Figures 14-17: I would suggest using black lines for the experimental curves.
- Given the differences in profiles before the hump, it is suppressing that Lund and RAND methods give so close results. How is it justified?
References
Aboshosha, H., Elshaer, A., Bitsuamlak, G.T. & El Damatty, A. (2015). Consistent inflow turbulence generator for LES evaluation of wind-induced responses for tall buildings. Journal of Wind Engineering and Industrial Aerodynamics, 142, pp.198–216.
Bazdidi-Tehrani, F., Kiamansouri, M. & Jadidi, M. (2016). Inflow turbulence generation techniques for large eddy simulation of flow and dispersion around a model building in a turbulent atmospheric boundary layer. Journal of Building Performance Simulation, 9(6), pp.680–698. Retrieved from https://www.tandfonline.com/doi/full/10.1080/19401493.2016.1196729
Tolias, I.C., Koutsourakis, N., Hertwig, D., Efthimiou, G.C., Venetsanos, A.G. & Bartzis, J.G. (2018). Large Eddy Simulation study on the structure of turbulent flow in a complex city. Journal of Wind Engineering and Industrial Aerodynamics.
Vasaturo, R., Kalkman, I., Blocken, B. & van Wesemael, P.J.V. (2018). Large eddy simulation of the neutral atmospheric boundary layer: performance evaluation of three inflow methods for terrains with different roughness. Journal of Wind Engineering and Industrial Aerodynamics, 173, pp.241–261.
Wu, X. (2017). Inflow Turbulence Generation Methods. Annual Review of Fluid Mechanics, 49(1), pp.23–49. Retrieved from http://www.annualreviews.org/doi/10.1146/annurev-fluid-010816-060322
Yan, B.W. & Li, Q.S. (2015). Inflow turbulence generation methods with large eddy simulation for wind effects on tall buildings. Computers and Fluids, 116, pp.158–175.
Yang, Q., Zhou, T., Yan, B., Van Phuc, P. & Hu, W. (2020). LES study of turbulent flow fields over hilly terrains — Comparisons of inflow turbulence generation methods and SGS models. Journal of Wind Engineering and Industrial Aerodynamics, 204, p.104230.
Author Response
Thanks for your kind suggestions about this paper. For the response, please see the attachment

Reviewer 2 Report
Dear authors, thank you for implementing my previous comments. The manuscript is much improved. Most of the remaining comments related to clarifications and providing more details. I vote for a minor revision.
Major Comments:
1- The paper is much improved since last I reviewed it. The English is nicely edited, and the objectives are set clearly.
2- Due to lack of time and expertise, I did not carefully check all the equations and text. Perhaps another reviewer can help in this regard. However, I checked the figures and tables.
3- The domain width is still small ~ 0.2 times of cord length. This essentially makes the simulation a 2D simulation with overlooking eddies and their effects on transport along the spanwise direction. To simulate those eddies, the spanwise domain should be at least ~ 1-2 times of cord length. This limitation should clearly be stated in the paper. I am glad that the authors mentioned this in the last paragraph of the conclusions. I encourage authors to also mention this up front when they define their objectives or development of methodology.
4- In my earlier comment, I said that the paper needs to disclose all the input parameters, constants, and variables, for each of the methods selected. This helps other researchers to recreated the simulations and compare their results to this paper. However, the authors still have not provided all the detailed settings for each of the inlet boundary methods. Please provide a detail table, listing all input variables, constants, etc. used for each of the inlet methods.
5- To ensure enough of the TKE is simulated (rather than modelled), can authors calculate the ratio of TKE simulated over the total TKE as a function of distance from the wall? This ratio should be close to 80% in the interior of the domain. This warrants that the LES mesh has sufficient resolution. Refer to the following reference:
Turbulent Flows, Stephen B. Pope, Cambridge University Press, 2000
Minor Comments:
1- For units please use negative exponents throughout: m/s -> m s^-1
2- Figures 12 etc. the first pannel (a) should lable the y axis and the U+ axis. These axes are currently missing. Same comment for Figs. 13, 14, 15, etc.
3- For calculation of y+ and U+ it is necessary to estimate the friction velocity u*. Can authors carefully specify how they calculated friction velocity? did they calculate it using flow fluctuations? and where? If they used some sort of log-law, they should also specify where they considered the log-law...
4- Fig. 17. the Reynolds stress component uv should be negative as it is correctly represented in panels b-e. However, panel (a) is unclear whether it is showing uv or - uv. Authors should clearly label the x-y axes for panel (a).
Author Response
Thanks for your kind suggestion about my work. For the response, please see the attachment.
